# scPower accelerates and optimizes the design of multi-sample single cell transcriptomic studies

Katharina T. Schmid [1,2], Barbara Höllbacher [1,2], Cristiana Cruceanu[3], Anika Böttcher [4,5,6],
Heiko Lickert [4,5,6], Elisabeth B. Binder [3,7], Fabian J. Theis [1,8] & Matthias Heinig [1,2 ✉]

Single cell RNA-seq has revolutionized transcriptomics by providing cell type resolution for differential gene expression and expression quantitative trait loci (eQTL) analyses. However, efficient power analysis methods for single cell data and inter-individual comparisons are lacking. Here, we present scPower; a statistical framework for the design and power analysis of multi-sample single cell transcriptomic experiments. We modelled the relationship between sample size, the number of cells per individual, sequencing depth, and the power of detecting differentially expressed genes within cell types. We systematically evaluated these optimal parameter combinations for several single cell profiling platforms, and generated broad recommendations. In general, shallow sequencing of high numbers of cells leads to higher overall power than deep sequencing of fewer cells. The model, including priors, is implemented as an R package and is accessible as a web tool. scPower is a highly customizable tool that experimentalists can use to quickly compare a multitude of experimental designs and optimize for a limited budget.

[1] Institute of Computational Biology, Helmholtz Zentrum München – German Research Center for Environmental Health, Neuherberg, Germany.
[2] Department of Informatics, Technical University Munich, Munich, Germany. [3] Department of Translational Research, Max Planck Institute for Psychiatry, Munich, Germany. [4] Institute of Diabetes and Regeneration Research, Helmholtz Diabetes Center, Helmholtz Zentrum München – German Research Center for Environmental Health, Neuherberg, Germany. [5] German Center for Diabetes Research (DZD), Neuherberg, Germany. [6] School of Medicine, Technical University of Munich, Munich, Germany. [7] Department of Psychiatry and Behavioral Sciences, Emory University School of Medicine, Georgia, USA.
[8] Department of Mathematics, Technical University Munich, Munich, Germany. ✉email: matthias.heinig@helmholtz-muenchen.de

Understanding the molecular basis of phenotypic variation, such as disease susceptibility, is a key goal of contemporary biomedical research. To this end, researchers use transcriptomic profiling to identify changes of gene expression levels (differentially expressed genes; DEGs) between sets of samples, e.g., patients and healthy controls[1–5]. Combining this with genetic information leads to the analysis of differential expression between genotypes and the identification of expression quantitative trait loci (eQTLs)[6–9], supplying the molecular link between genome and phenotype[10].

Single cell RNA-sequencing (scRNA-seq)[11–15] allows for differential gene expression and eQTL analysis on the level of individual cell types. Typically, single cell differential gene expression analysis seeks to identify genes whose expression levels are markedly different between different cell types[16–18]. In contrast, multi-sample experiments aim at the identification of DEGs between sets of samples within the same cell type. These sets can be defined by different experimental conditions or genotypes and are each measured at the single cell level. Multi-sample experiments have been identified as one of the grand challenges for single cell data analysis[19].

Power analysis is an important step in the design of statistically powerful experiments given certain assumptions about the expected effect sizes and constraints on the available resources. Researchers need to decide on parameters such as the sample size, the number of cells per sample and the number of reads. The power is tightly linked with the statistical testing procedure. Several methods have been established based on the theory of linear regression models[20] and the control of the false discovery rate[21–24] for microarray studies. For bulk RNA-seq studies, power analysis methods based on the theory of negative binomial count regression[25,26], other parametric models[27–29], or simulations[30,31] have been proposed and benchmarked[32].

In principle methods for bulk RNA-seq power analysis could also be applied to compute power or minimally required sample sizes for given effect sizes for single cell experiments, however they fail to take into account specific characteristics of single cell data. In scRNA-seq experiments, individual cells are typically not sequenced to saturation, leading to sparse count matrices, where only highly expressed genes are detected with counts greater than zero. In addition, the overall number of transcripts as well as the number of transcripts of individual genes can be highly cell type specific[33].

Recently, individual aspects of single-cell specific experimental design were addressed (Supplementary Table S1). First, recommendations of sequencing depth have been obtained by comparing sensitivity and accuracy of different technology platforms[34–36]. Second, it has been established that the minimal number of sequenced cells required to observe a rare cell type with a certain frequency can be modelled with a negative binomial distribution[37,38] or multinomial distribution[39].

While these insights also help with the design of multi-sample experiments, there are additional parameters that need to be taken into account such as the sample size and the effect sizes. For single cell differential expression analyses, several simulation-based methods have been published recently which estimate the power dependent on the effect size between the groups[40–43]. However, only one simulation tool also addresses multi-sample comparisons with cells from different individuals in each group and can thereby give recommendations for the sample size[43].

Two benchmarking studies, one applying the aforementioned simulation tool and one using different example data sets, demonstrated that the "pseudobulk" approach in combination with classical differential gene expression methods such as edgeR[44] and limma-voom[45] outperforms single cell specific methods and mixed models in multi sample DE analysis[43,46]. The pseudobulk approach approximates cell type specific gene expression levels for each individual as the sum of UMI counts over all cells of the cell type and was also successfully applied in different single cell eQTL studies[47–49].

While simulations[43] successfully assess the power of the pseudobulk approach, they suffer from a number of shortcomings. A big disadvantage of simulation-based studies are their long runtimes, which make them unsuitable to evaluate the large number of experimental designs needed to optimize parameter combinations. Even power analysis for a single experiment with a large sample size can be very memory and runtime intensive. In addition, handling more complex designs is not easily accomplished with simulation based methods, but could be achieved with analytical power analysis methods.

A first analytic exploration of different experimental designs for single cell eQTL studies showed the importance of the optimizing parameters for a restricted budget[50], as shallow sequencing of more samples can increase the effective sample size. However, the analysis provided no generalizable tool that can be applied on other data sets and is missing an exact power estimation based on effect size priors. Furthermore, it is not applicable for DEG analysis.

Here, we provide a resource that enables choosing the optimal experimental design for interindividual comparisons. It focuses on the power to detect DEGs and eQTLs while also addressing the power to detect rare cell types. Our model was specifically developed for the pseudobulk approach, including a quantification of the probability to detect cell type specific gene expression in scRNA-seq data. We ensure an accurate power estimation with our model by selecting appropriate priors for the cell type specific expression distributions and for the effect size distributions. We derive data driven priors on expression distributions from single cell atlases of three different tissues[51,52]. We combine these with cell type specific priors for effect sizes based on DEGs and eQTL genes from bulk RNA-seq experiments on cells sorted by fluorescence activated cell sorting (FACS)[53–57]. Comparing our method against established simulation-based approaches validates our power estimates. In contrast to simulation-based methods, our analytic method can efficiently test a multitude of design options, making it suitable for the optimization of experimental parameters. Our model provides the basis for rationally designing well powered experiments, increasing the number of true biological findings and reducing the number of false negatives.

Efficient calculation including a selection of different possible priors is easily accessible for the user, as we provide our model and parameters as an open source R package scPower on github https://github.com/heiniglab/scPower. All code to reproduce the figures of the paper is provided in the package vignette. The repository includes a shiny app with a user-friendly graphical user interface, which is additionally available as a web server at http://scpower.helmholtz-muenchen.de/.

## Results

**Power analysis framework for scRNA-seq experimental design.** Our power analysis framework targets multi-sample transcriptomic experiments analyzed with the pseudobulk approach. Each analysis starts with a count matrix of genes times cells. These counts can either be counts of unique molecular identifiers (UMI), in the case of droplet-based technologies, or read counts in the case of Smart-Seq. Cells are annotated to an individual and a discrete cell type or state. These can be derived by clustering and analysis of marker genes, potentially considering multiple levels of resolution[58] or using the metacell approach[59]. Individuals are annotated with different experimental covariates, such as disease

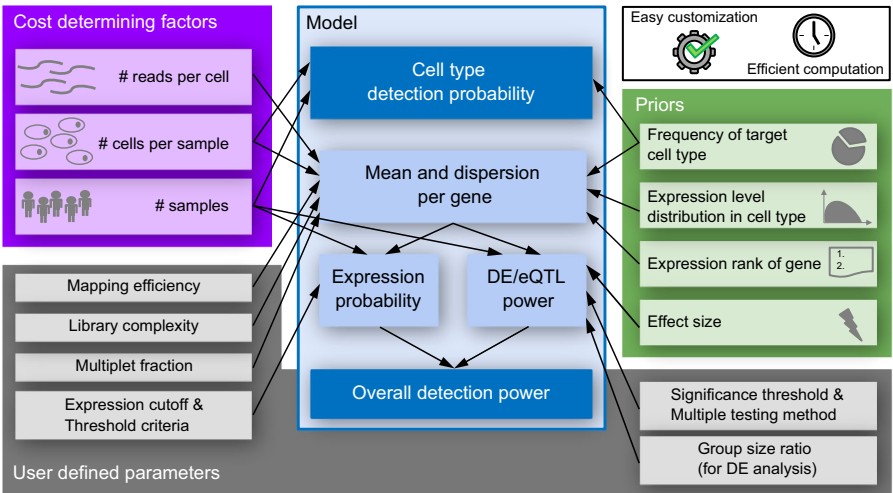

**Fig. 1 Dependence of experimental design parameters.** The cost determining factors (purple: number of samples, number of cells per sample and number of reads per cell) affect the overall detection power through the expression probability and the DE/eQTL power (blue). In addition, the power depends on prior knowledge or assumptions (green) as well as user-defined parameters such as the significance threshold and the expression cutoff (grey). Our model enables fast power calculation, independent of the chosen experimental parameters, and easy adaptation to different use-cases through reference priors.

status. We focused on two group comparisons, but more complex experimental designs, which can be analyzed with generalized linear models, can also be accommodated (see package vignette).

To determine cell type specific differential expression between samples, gene expression estimates for each sample and each cell type are approximated as the sum of (UMI) counts over all cells of the cell type[47–49]. This pseudobulk approach has been identified as one of the currently best performing approaches for multi sample DE analysis in recent benchmarking studies[43,46]. It is important to keep in mind that the pseudobulk approach on single cell data is distinct from traditional bulk RNA-seq. In pseudobulk the ability to detect the expression of a gene depends on the number of cells of the cell type and on the expression level of the specific gene. Therefore, we model the general detection power dependent on the number of cells per sample $n_c$ which is related to the number of cells per cell type. Two additional experimental parameters determine the power in our model and also the cost of a scRNA-seq experiment in general: the number of samples $n_s$ and the number of reads sequenced per cell $r$. In order to compute the power of the experiment, we either need to make explicit assumptions or use prior knowledge about unknown experimental parameters, such as the assumed effect sizes and gene expression levels of eQTLs and DEGs. This prior knowledge is combined with user-defined parameters and cost determining factors to model the overall detection power (Fig. 1).

In order to choose the optimal parameter combination of sample size, cells per sample and read depth for an experimental design, there are two types of power to consider. First, the power to detect the cell type of interest and second, the power to detect DE/eQTL genes within this cell type (i.e., overall detection power).

The power to observe the cell type of interest depends on its frequency, the number of cells sequenced per individual and the total number of individuals. Following Abrams et al.[37], we model this problem using the negative binomial distribution (see "Methods"). Using prior knowledge of cell proportions in peripheral blood mononuclear cells (PBMCs) from the literature, we determine the number of cells required for each individual to detect a minimal number of cells of a specific type (Supplementary Fig. S1). The comparison for varying numbers of individuals shows that the number of cells required for each individual is

most strongly affected by the frequency of the cell type and only to a smaller degree by the number of individuals.

The power to detect DE/eQTL genes within this cell type is called overall detection power $P$. Our framework models $P$ of an experiment across all considered DEGs/eQTL genes $D$ conditional on the experimental design parameters and the priors. The overall detection power is defined as the mean gene level detection power $P_i$ conditional on gene specific priors of gene $i$:

$$P = \frac{1}{|D|} \sum_{i \in D} P_i \qquad (1)$$

In order to identify a gene as an DEG/eQTL gene, it must be both expressed and exceeding the significance cutoff. Therefore, we further decompose the gene level detection power $P_i$ into the expression probability $P(i \in E)$, which quantifies the probability to detect gene $i$ in the set of expressed genes $E$, and the DE/eQTL power, which we denote as the probability $P(i \in S)$ that gene $i$ is in the set of significant differentially expressed genes $S$. This quantifies the power (probability to reject $H_0$ when $H_1$ is true) of the statistical test for gene $i$ and depends on the assumed effect sizes $\Theta_p$, which can be derived from prior data, and the multiple testing adjusted significance threshold $\alpha$. In addition, both the expression probability and the DE/eQTL power depend on the mean $\mu$ and dispersion $\phi$ of expression levels of gene $i$. In our model $\mu$ and $\phi$ are determined by the experimental design parameters $(n_c, r)$ and the parameters of cell type specific expression distributions $\Theta_e$. Conditioning the gene level detection power $P_i$ on these priors and experimental design parameters, allows for decomposing $P_i$ as the product of the expression probability and the DE/eQTL power:

$$P_i = P(i \in E \wedge i \in S | n_s, n_c, r, \Theta_e, \Theta_p, \alpha) =$$
$$= P(i \in E | n_s, \mu(n_c, r, \Theta_e), \phi(n_c, r, \Theta_e)) \cdot \qquad (2)$$
$$P(i \in S | n_s, \mu(n_c, r, \Theta_e), \phi(n_c, r, \Theta_e), \Theta_p, \alpha)$$

In the following sections the models for the gene level expression probability and the DE/eQTL power are specified.

**scPower accurately models the number of detectable genes per cell type.** In scRNA-seq experiments, typically only highly expressed genes are detected with counts greater than zero[34–36].

This sparsity makes it difficult to assess gene expression levels and probabilities of detecting expressed genes of future experiments. We tackled this by modelling the cell type specific expression distribution based on the number of reads sequenced per cell $r$, the number of cells of the cell type per individual $n_{c,s}$ and the number of individuals $n_s$. Taken together with a user-defined cutoff, this allows us to accurately predict the number of detectable genes per cell type. In the following sections, we explain how we parameterize the model by these three variables.

In order to model expression probabilities that are cell type-specific, we need to take into account that the overall RNA abundance and distribution varies between different cell populations[33]. These cell type specific differences can be captured in priors that describe the general expression distribution in the target cell types. We illustrate our expression probability model and the strength of expression priors on various blood cell types. To this end, we fit the expression priors per cell type using a scRNA-seq data set of PBMCs from 14 healthy individuals measured with 10X Genomics (Supplementary Fig. S2, Table S2), in the following called the training data set, and evaluate it on a second independent PBMC data set[47], the validation data set. Of note, the pilot data should in general represent controls without strong DE effects, that cover the natural inter-sample variability.

For the cell type specific expression prior, we approximate the single cell count distribution in each cell type with a small number of hyperparameters dependent on the read depth (Fig. 2a). We model UMI counts per gene $i$ in a particular cell type $c$ as independent and identically distributed according to a negative binomial distribution with a mean $\mu_{i,c}$ and dispersion parameter $\phi_{i,c}$. The distribution of means $\mu_{i,c}$ across all genes is further modeled as a mixture distribution with a zero component and two left censored gamma distributions to cover highly expressed genes (see "Methods" and Supplementary Fig. S3). Subsampling the read depth of our data shows that the parameters of the mixture distribution are linearly dependent on the average UMI counts (Supplementary Fig. S4). The dispersion parameter $\phi_{i,c}$ is modelled dependent on the mean $\mu_{i,c}$, using the approach of DEseq[60]. As the initial experimental parameter for our model is the read depth and not directly the UMI counts, average UMI counts are related to the average number of reads mapped confidently to the transcriptome, which are in turn related to the number of reads sequenced per cell (Supplementary Fig. S5).

Taken together, we now have a model of per cell read counts across all genes parameterized by the number of reads sequenced, which was trained on cell type specific expression data. The set of parameters describing the gamma mixture distribution dependent on the UMI counts, the mean-dispersion curves and the read depth-UMI curves is called expression prior in the following. It is required for a correct modelling of the count distribution in unseen data and so the expression probabilities. We provide expression priors for 25 different cell types from 3 different tissues in scPower and the user can easily generate their own expression priors for missing cell types with our package.

We can now use these expression priors to quantify the expression probability of all genes in a future experiment with different experimental parameters. For this, we quantify the expression distribution of a particular gene in a particular cell type and individual based on its prior expression strength. This prior is represented by the expression rank of the gene compared to all other genes. We determine its mean expression level as the quantile corresponding to this expression rank in the single cell expression prior distribution. This quantity is dependent on the read depth. Next, we derive the pseudobulk count distribution from the single cell expression distributions. This pseudobulk count distribution is again a negative binomial distribution. Its mean and dispersion are scaled by the number of cells per individual and cell type.

Whether a gene is expressed or not, can now be estimated based on this gene specific pseudobulk distribution, combined with a user defined threshold. In our default settings, the threshold is composed of a minimum pseudobulk count (sum of UMI counts per gene per cell type per individual) and a certain fraction of individuals. Specifically, we compute the probability that the observed counts are greater than the user defined minimal count threshold in at least a given number of individuals. Summing up these gene expression probabilities allows for modelling the expected number of expressed genes (see "Methods" section for detailed formulas). On top of our default threshold criteria, our package offers the user alternative options for expression thresholds, e.g., that a gene is called expressed if it has a count > 0 in a certain percentage of cells.

Subsampling of our data shows that the number of expressed genes per cell type depends on the number of cells of the cell type and the read depth (Fig. 2b, c). The observed numbers of expressed genes (solid lines) are closely matched by the expectation under our model (dashed lines), shown here with example cutoffs of counts greater than ten and zero. We show the results for one batch of the PBMC data set (Fig. 2b), while the fits of all batches can be found in Supplementary Figs. S6 and S7. Predicted and observed numbers of expressed genes were highly correlated (all $r^2 > 0.9$, Supplementary Table S3).

To validate our model, we applied it on a second PBMC data set[47] that was not used during parameter estimation for the expression priors (Fig. 2c). This validation data set was measured at a smaller read depth of 25,000 reads per cell and for a different sample size (batch A and B with 4 individuals and batch C with 8 individuals). The observed numbers are closely matched by the expectation under our model (all $r^2 > 0.9$), which demonstrates that it can generalize well between data sets and different experimental parameters. Taken together, we now have a general model for the expected number of expressed genes, which is parameterized by the number of cells per cell type and the number of reads per cell. Of note, gene expression distributions are cell type specific and the model parameters have to be fitted from suitable (pilot) experiments, such as the human cell atlas project[61].

**scPower models the power to detect differentially expressed genes and expression quantitative trait genes.** Building on our expression probability model, we can assess the DE/eQTL power of the expressed genes using existing analytical power analysis tools that have been established for bulk sequencing data. They estimate the power to detect an effect of a given effect size depending on the sample size, the gene mean expression level and the chosen significance threshold. Analytic power analysis compares the distributions of the test statistic under the null and the alternative model (e.g., applying a certain effect size). Based on the significance threshold the critical value of the test statistic is determined from the null distribution. Then the power is given by the probability mass of the distribution under the alternative model that exceeds the critical value.

An adjustment of the significance threshold is necessary due to the large number of parallel tests performed in a DEG analysis in order to avoid large numbers of false positive results. We provide two methods in our framework for that, either controlling the family-wise error rate (FWER) using the Bonferroni method[62] or the false discovery rate (FDR)[22]. In the following analyses, we used the FDR adjustment for DE power and FWER adjustment for eQTL power, as proposed by the GTEx Consortium[63] for a genome-wide *cis* eQTL analysis.

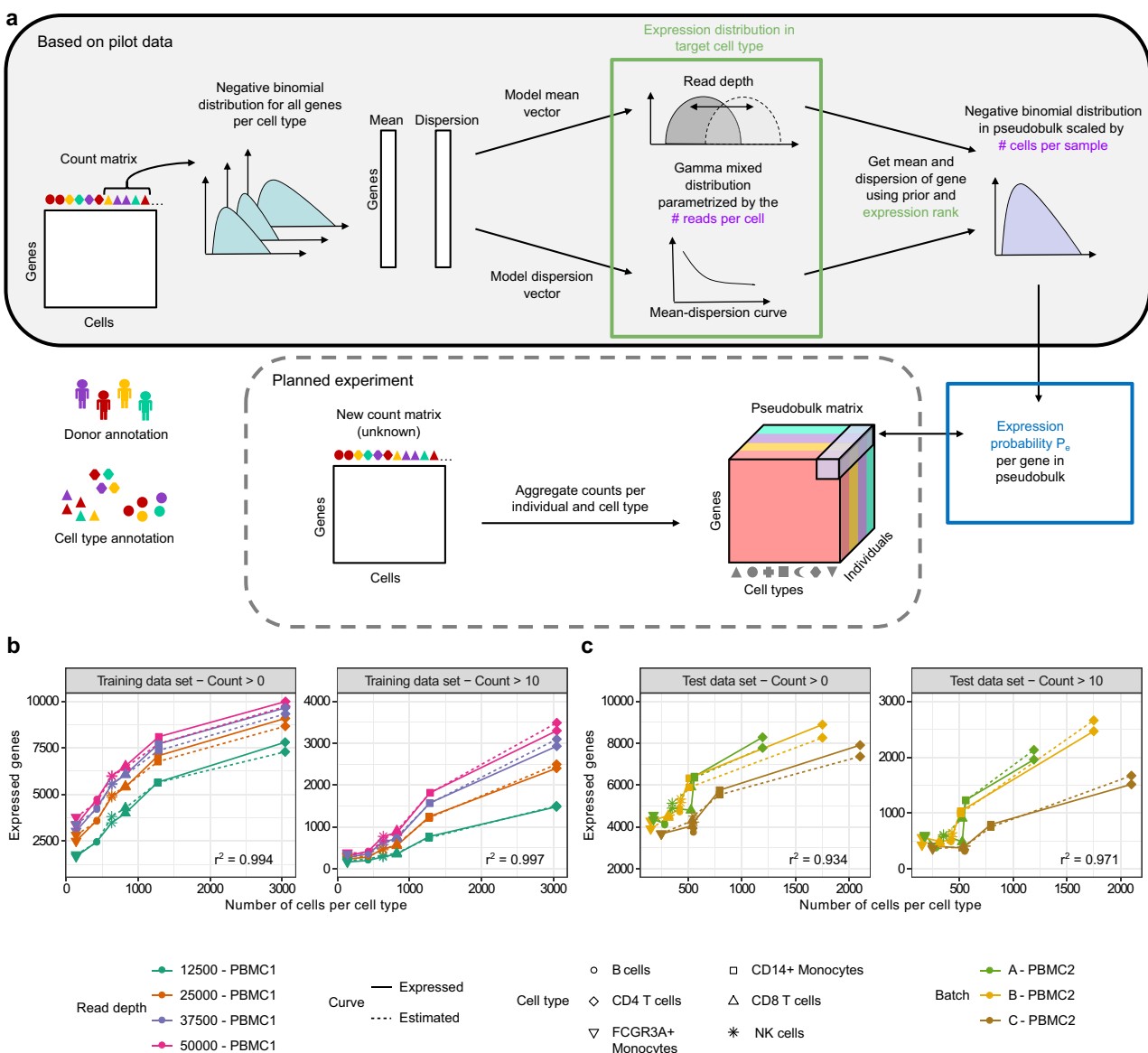

**Fig. 2 Expression probability model parameterized by UMI counts per cell. a** The expression probabilities for genes in pseudobulk of a newly planned experiment are estimated based on the expression prior and the planned experimental parameters. For this, the expression prior is derived from the mean and dispersion parameters of gene-wise negative binomial distributions fitted from a matching pilot data set. **b** Using this approach, the number of expressed genes expected under our model (dashed line) closely matches the observed number of expressed genes (solid line) dependent on the number of cells per cell type (cell type indicated by point symbol) for one batch of the training PBMC data set (Supplementary Table S2). The data is subsampled to different read depths (indicated by colour). The r² values between estimated and expressed genes were highly significant for both expression thresholds. **c** The model performed similarly well for the three batches of an independent validation PBMC data set[47]. Used expression threshold: count > 10 (right panels of **b**, **c**) or count > 0 (left panels of **b**, **c**) in more than 50% of the individuals.

Specifically, the power analysis methods we apply for DE and eQTL studies are based on negative binomial regressions[64] and linear regressions[20], respectively. This also leads to different effect size specifications; fold changes in the DE case and R-squared values in the eQTL case. Of note, the R-squared values combine allele frequency and beta value in the linear model.

For DE analysis, power calculations are based on negative binomial regression, which is a powerful approach used in tools such as DESeq[5,60] or edgeR[44] for DEG analysis of both RNA-seq and scRNA-seq[18,65–67]. Benchmarking studies showed that these tools combined with the pseudobulk approach outperform other methods in multi-sample differential expression analysis[43,46]. We

verified that all our training data sets could be modelled by negative binomial distributions after pseudobulk transformation and found no evidence of zero inflation (Supplementary Table S4). In contrast to the other technologies, the Smart-seq2 data showed zero-inflation on the single cell level (see also[68,69]), but aggregation to pseudobulk removed the excess of zero values. Hence, it is valid to apply analytical methods for the power analysis of negative binomial regression models[64]. To obtain a range of typical effect sizes and mean expression distributions in specific immune cell types, we analyzed several DEG studies based on FACS sorted bulk RNA-seq[53,54] (Supplementary Figs. S8 and S9). Combined with our gene expression model, we can

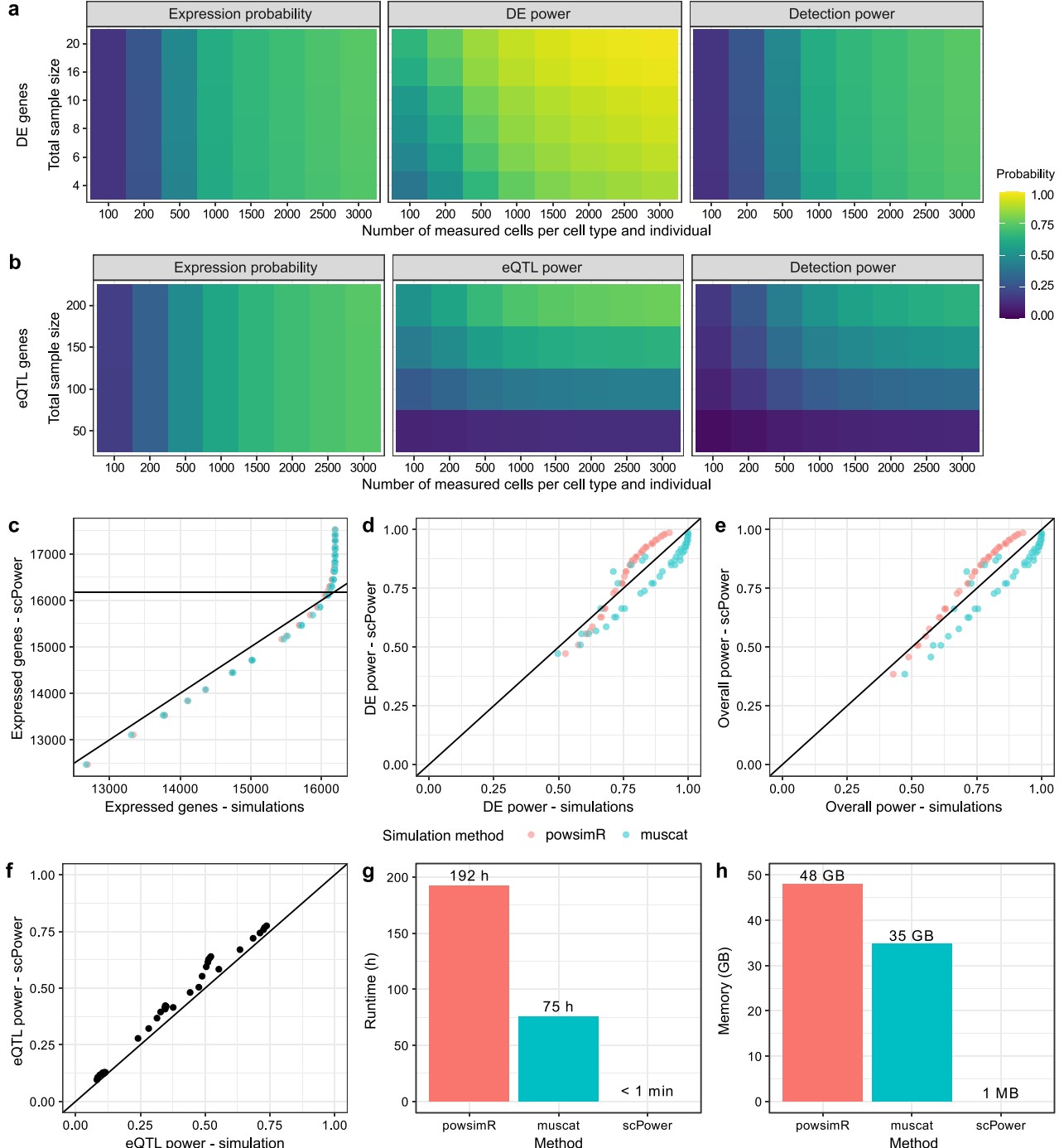

**Fig. 3 Expression probability, DE/eQTL power and overall detection power and their validation in simulation studies.** Power estimation using data driven priors for DE genes (**a**) and eQTL genes (**b**) dependent on the total sample size and the number of measured cells per cell type. The detection power is the product of the expression probability and the power to detect the genes as DE or eQTL genes, respectively. The fold change for DEGs and the $R^2$ for eQTL genes were taken from published studies, together with the expression rank of the genes. For (**a**), the Blueprint CLL study with comparison iCLL vs mCLL was used, for (**b**), the Blueprint T cell study. The expression profile and expression probabilities in a single cell experiment with a specific number of samples and measured cells was estimated using our expression prior, setting the definition for expressed to > 10 counts in more than 50% of the individuals. Multiple testing correction was performed by using FDR adjusted $p$ values for DE power and FWER adjusted $p$ values for eQTL power. The probabilities calculated in (**a**) were verified by the simulation-based methods powsimR and muscat with each point representing one parameter combination. **f** The eQTL power of (**b**) could be replicated with a self-implemented simulation. Runtime (**g**) and memory requirements (**h**) were drastically higher in the simulations than for our tool scPower during the evaluations of (**c–e**), showing the strength of our analytic model.

calculate the overall detection power of DE genes averaging over the gene specific expression probability times the power to detect the gene as a DE gene based on fold changes from prior DEG studies. In the following analyses, we assume a balanced number of samples for both groups, but scPower can also evaluate unbalanced comparisons, which lead to a decrease in power.

Using fold changes from a study comparing CLL subtypes iCLL vs mCLL[53] as effect size priors (sample size of 6, 84 DEGs with median absolute log fold change of 2.8) we find a maximum overall detection power of 74% (Fig. 3a). This power is reached with the experimental parameters of 3000 cells per cell type and individual, a total balanced sample size of 20, i.e., 10 individuals per group, and FDR adjusted $p$ values. For this parameter combination and prior, the DE power would reach even 98% for all DE genes of the study, however, only 74% are likely to be expressed. Overall, the DE power increases with higher number of measured cells and higher sample sizes, while the expression probability is mainly influenced by the number of measured cells.

The influence of the sample size is not so pronounced in this example due to the small sample size of the reference study. Potential weaker effect sizes that would be identified with larger sample sizes could not be considered in the priors, which leads to a low required sample size for the power estimation. For other reference studies the impact of a higher sample size on the power is more visible (Supplementary Fig. S10). Similar detection ranges are found for the comparison of other CLL subtypes in the same study, while the detection power in a study of systemic sclerosis vs control were much lower with values up to 30% (Supplementary Fig. S10). Smaller absolute fold changes in this study decrease the DE power and therefore also the overall detection power. The effect of using the FWER adjustment also for the DE power can be seen in Supplementary Fig. S11.

For eQTL analysis, power calculations are based on linear models[20]. Due to the very large number of statistical tests (~millions), simple linear models are usually applied to transformed read count data[45,70], as they can be computed very efficiently. For large mean values, the power is estimated analytically, for small mean values, this approximation can be imprecise and instead simulations are used that take the discrete nature of scRNA-seq into account. This introduces a dependency between the eQTL power and the expression mean and thus eQTL power is considered conditional on the mean. The mean threshold below which simulations are used, was defined by comparison of simulated and analytic power (Supplementary Fig. S12).

Overall detection power for eQTL genes (Fig. 3b) shows a stronger effect of the sample size, which increases the eQTL power. In the depicted use case, the applied priors originate from an eQTL study of T cells from the Blueprint consortium[57], which had a sample size of 192 and identified 5,132 eQTL genes with a median absolute beta value for the strongest associated SNP of 0.89. Increasing the number of cells per individual increases both the expression probability and the eQTL power by shifting the expression mean of the pseudo bulk counts to higher values. Notably, increasing the number of measured cells per individual and increasing the sample size both result in higher costs. A maximal detection power of 64% was found for a sample size of 200 individuals and 3,000 measured cells per cell type and individual. The Blueprint eQTL data set also contains eQTLs from monocytes where we observe the same trend and found a maximal detection power of 65% (Supplementary Fig. S11).

**scPower estimations are supported by simulations**. The accuracy of scPower was evaluated by benchmarking against different simulation-based methods (Fig. 3c–f). In general, simulation-based methods generate and analyze example count matrices.

Therefore, they are always approximations and need to be repeated multiple times for accurate results, while we transformatively enable the design of experiments with our analytic model that requires order of magnitude less runtime and memory (Fig. 3g-h).

For single cell DE experiments, we compared our model with powsimR[40] and muscat[43], which both show well matching power estimations compared to our tool scPower. powsimR is a widely used simulation-based method that is however not designed for multi-sample single cell comparison, i.e., it is only possible to make comparisons of groups of single cell measurements within the same sample but not between multiple samples. Adaptations of powsimR were necessary to make it comparable to scPower (see "Methods" for a detailed description of changes). In contrast, muscat is a recent method that already incorporates the pseudobulk approach for multi-sample comparison and can be used directly. Both simulation methods can be combined with different DE analysis methods for the downstream analysis of the simulated counts. We evaluated them in combination with different common DE methods, such as DESeq2[5], edgeR[44] and limma[45].

The simulation based power estimates from the adapted version of powsimR as well as from muscat matched the estimates from scPower very well (Fig. 3c–e). We compared the expected number of expressed genes, the DE power of these expressed genes and the overall power for all simulated genes. Running simulations with different DE methods showed that the observed power also depends on analysis choices such as the DE method with scPower estimates being most accurate when using edgeR (Supplementary Fig. S13). Furthermore, powsimR and muscat differ slightly, caused by different modelling assumptions. The overall trends when comparing different experimental designs are in good agreement between scPower and all analysis methods applied to the simulated reads. This is true for both FWER adjustment and FDR adjustment as multiple testing correction. A comparison over a wide range of experimental design parameters between edgeR applied to simulated data from powsimR and scPower confirms the agreement of power estimates (Fig. 3c–e and Supplementary Fig. S14).

Furthermore, we used the simulation-based methods to evaluate how well our power analysis method performs for different real-life conditions, such as batch effects or unbalanced cell type proportions between the groups. Simulating batch effects showed a clear drop in power, especially if the magnitude of the batch effect is larger than the effect size of DEGs (Supplementary Fig. S15). However, under the assumptions of an unconfounded experimental design with batches containing both controls and cases, batch effects can be removed by adding a batch covariate to the regression model[71]. This increases the power compared to non-batch corrected analyses[72–74]. Following this strategy, we could recover the same power as in experiments without batch effects, i.e., our power estimations stay accurate in experiments with batch effects, given that they are adjusted for in the analysis.

A second source that can lead to a reduction of power are different cell type proportions in both groups (Supplementary Fig. S16). In this case, a conservative power estimation can be achieved by setting the expected cell type frequency to the frequency of the smaller group. This represents a good lower bound estimation, especially in cases with small sample sizes.

Contrary to DE analysis, there currently exists no power estimation method for single cell eQTL that explicitly accounts for specific effect size priors. Therefore, we compared the analytical eQTL power with our own simulation method, which is also used for power estimation of genes with small mean values. The simulation method applies our underlying expression probability model of scPower for assigning a mean value to each gene. This part of the model is the same for eQTL and DE power

and was already shown to be accurate compared to powsimR and muscat. Therefore, we focus on benchmarking the eQTL power, which showed good agreement between the simulated and analytic values (Fig. 3f).

The analytic calculations of scPower are orders of magnitude faster than the simulation-based approaches: calculations for Fig. 3c–e took 8 days for powsimR, 3 days for muscat and less than a minute for scPower (Fig. 3g). Also the memory requirements are much lower, as no count matrices are generated. For the simulation-based methods the memory requirements increase with larger sample size and numbers of cells, leading for example for 20 samples and 3000 cells per sample to 48GB used memory for powsimR and 35GB used memory for muscat compared to the parameter-independent requirements of scPower of few MB (Fig. 3h). In addition the installation of scPower is easier due to less dependencies: 11 dependencies of scPower vs. 82 dependencies of powsimR and 28 dependencies of muscat. These advantages of scPower over simulation based approaches enable a systematic evaluation of a large number of design options as described in the next section.

**scPower maximizes detection power for a fixed budget by optimizing experimental parameters.** With this model for power estimation in DE and eQTL single cell studies in place, we are now able to optimize the experimental design for a fixed budget. The overall cost function for a 10X Genomics experiment is the sum of the library preparation cost and the sequencing cost (see "Methods"). The library preparation cost is defined by the number of measured samples and the number of measured cells per sample, while the sequencing cost is defined by the number of sequenced reads, which also depends on the target read depth per cell.

We evaluated exemplarily the three parameters maximizing detection power, given a fixed total budget (Fig. 4). In this scenario, the optimal parameter combinations are identified for a DE study with a budget of 10,000€ (Fig. 4a) and for an eQTL study with a budget of 30,000€ (Fig. 4b). Besides the budget, the user can choose a criterion and threshold to define whether a gene is expressed. We followed the recommendation of edgeR[75] that the expression cutoff should correspond to the percentage of samples in the smaller group. For our DE example, this results in a percentage threshold of 50% due to the balanced DE design. For the eQTL example, we consider an eQTL with a minor allele frequency of 0.05, which is a common lower threshold for genetic variants tested for associations. We suggest that the gene should be at least expressed in the heterozygotes and thus pick a percentage threshold of 9.5% (see "Methods").

We use our method to calculate the overall detection power for different parameter combinations of cells per individual and read depth, while the sample size is defined uniquely given the other parameters and the fixed experimental budget. For the DE study with this specific prior combination, the optimal parameters are measuring 1200 cells in 4 samples with a read depth of 30,000. Measuring more cells per individual increases the expression probability and so the overall detection power (Fig. 4c), but due to the fixed budget this goes hand in hand with measuring less samples which decreases the DE power. A similar trend exists for the read depth (Fig. 4d).

For the eQTL study with this specific prior combination, the optimal parameters are measuring 1500 cells in 242 samples with a read depth of 10,000. Again a balance of the eQTL power, which depends mostly on the sample size, and the expression probability, which depends mostly on the cells per sample and the read depth, is visible (Fig. 4e–f). A user-specific version of this

analysis with custom budget and priors can be generated using our webtool http://scpower.helmholtz-muenchen.de.

We can expand our analyses with expression priors from our 10X PBMC data set and find the optimal parameter combinations depending on a given experimental budget (Fig. 5). We systematically investigated the evolution of optimal parameters for increasing budgets in four prototypic scenarios for DEG (Fig. 5a) and eQTL analysis (Fig. 5b), four scenarios based on prior DEG (Fig. 5c) and two scenarios on prior eQTL (Fig. 5d) experiments on FACS sorted cells (for the estimated costs see Supplementary Table S5). The prototypic scenarios reflect combinations of effect sizes (high, low) and expression ranks (high, low) of DEGs and eQTL genes. We observed that the number of cells per individual is the major determinant of power, as this is the variable that is either directly set to maximum values or increased first in the optimization (Fig. 5). This effect is least pronounced in the prototypic eQTL scenario (Fig. 5b), where small effect sizes require large sample sizes. For most DEG scenarios, the number of reads per cell is increased before increasing the sample size (Fig. 5a,c), indicating that strong effects can be detected with relatively few samples, while the detection of expression requires deeper sequencing. For eQTL scenarios, increasing the sample size first is more beneficial than increasing the read depth (Fig. 5b,d), which remains relatively low (10,000 reads per cell).

Figure 5 was generated with FDR adjusted p values for DE power and FWER adjusted p values for eQTL power. Using FWER adjustment for DE power changes the observed overall power, but leads to very similar optimal parameter combinations and the same trends overall (Supplementary Fig. S17).

In the cost optimization, we also took into account that increasing the number of cells per lane leads to higher numbers of doublets, i.e., droplets with two instead of one cell. Doublet detection methods such as Demuxlet[47] and Scrublet[76] enable faithful detection of those to exclude the doublets from the downstream analysis. We validated the doublet detection and donor identification of Demuxlet using our PBMC data set by comparing the expression of sex specific genes with the sex of the assigned donor (Supplementary Fig. S2b) and found high concordance after doublet removal, also for run 5, which was overloaded with 25,000 cells.

The increase of the doublet rate through overloading was modeled using experimental data[77] to accurately estimate the number of usable cells for the eQTL/DEG analysis. However, we observe in our own data set as well as in published studies[47,78] slightly higher doublet rates than shown in[77]. Therefore, we consider the modeled doublet rate as a lower bound estimation. With a high detection rate of doublets, overloading of lanes is highly beneficial, since larger numbers of cells per individual lead to an increase in detection power, while not causing additional library preparation costs. This supports previous evaluations that demonstrated the benefit of overloading[50]. Even though overloading leads to a decreasing number of usable cells and a decreasing read depth of the singlets, as doublets contain more reads, the overall detection power still rises strongly for both DE and eQTL studies.

**scPower generalizes across tissues and scRNAseq technologies.** Our power analysis framework is applicable to data sets for other tissues besides PBMC and to other single cell technologies besides 10X Genomics. We demonstrate this with a lung cell data set measured by Drop-seq[52] and a pancreas data set measured by Smart-seq2[51].

Drop-seq is a droplet-based technology similar to 10X Genomics, which is why we only need minor adjustments to our model. We set doublet rates as a constant factor, since Drop-seq does not provide information on the effect of overloading and

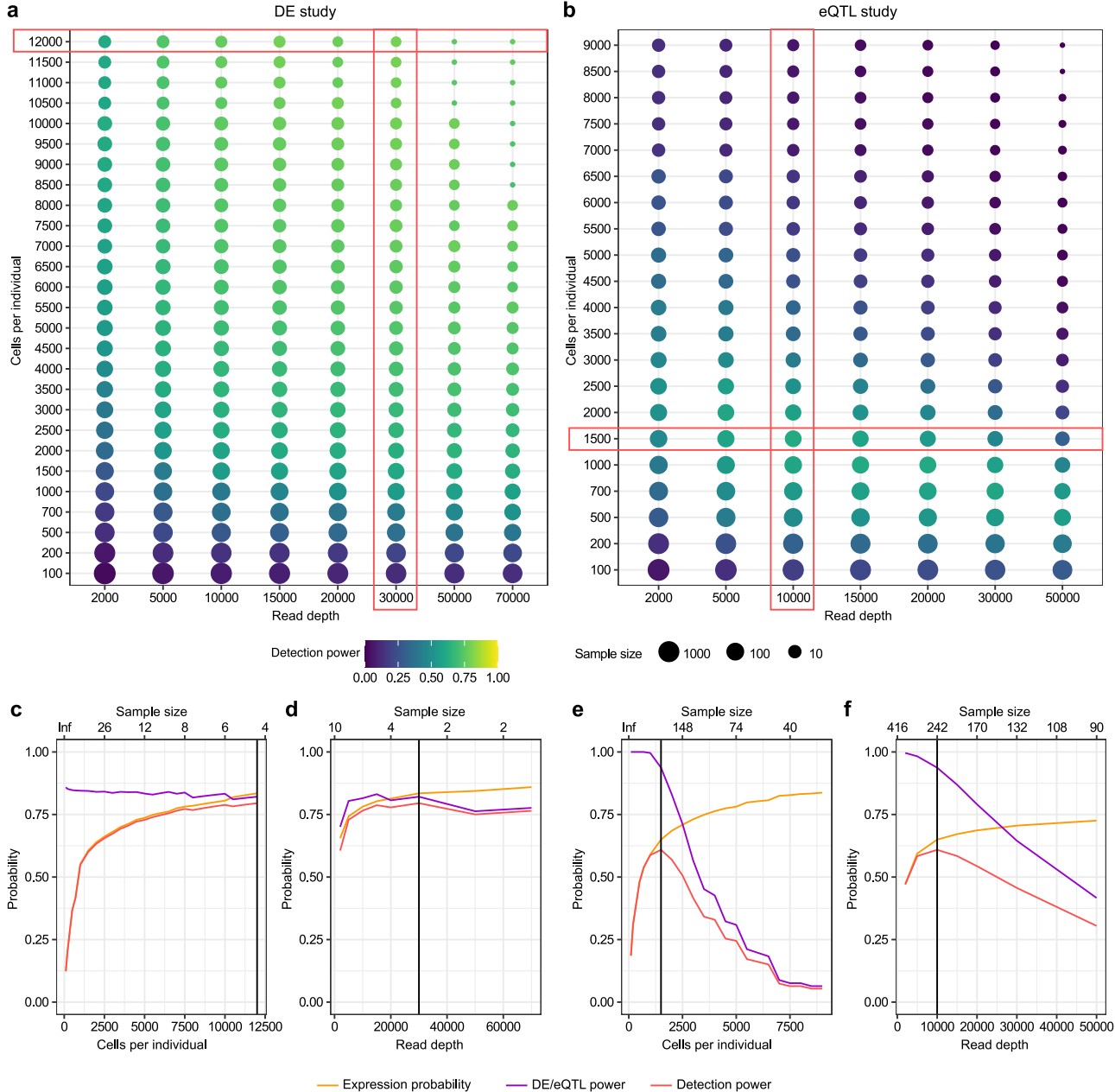

**Fig. 4 Parameter optimization for constant budget.** Maximizing detection power by selecting the best combination of cells per individual and read depth for a DE study with a budget of 10,000€ (**a**) and an eQTL study with a budget of 30,000€ (**b**). Sample size is uniquely defined given the other two parameters due to the budget restriction and visualized using the point size. **c–f** Overall detection power dependent on cost determining factors. Influence of the cells per individual given the optimized read depth (**c**, **e**) and of the read depth given the optimized number of cells per individual (**d**, **f**). Corresponds to the DE study in (**a**), visualized in (**a**) by the red frame around the row with the optimal number of cells (corresponding to (**c**)) and the red frame around the column with the optimal read depth (corresponding to (**d**)). Same frames for (**e**, **f**) in the eQTL study (**b**). The optimal sample size values are shown in the upper x axes for (**c–f**). Vertical line in the subplots marks the optimal parameter combination. Effect sizes were chosen as in Fig. 3. Gene expression is defined as detected in >50% (DE analysis) or >9.5% (eQTL analysis) of individuals.

from there, the DE/eQTL power calculations are the same as for 10X Genomics.

Smart-seq2 is a plate-based technology, generating read counts from full-length transcripts. To correct for the resulting gene-length bias, we express the count threshold for an expressed gene relative to one kilobase of the transcript. We fitted the expression model including the transcript length in the size normalization factor of the count model. In addition, as the technology is sorting individual cells into wells and does not suffer from variable doublet rates due to overloading, we modelled the doublet rate as a constant factor.

With these adaptations, our expression probability model (Supplementary Fig. S18) for both Drop-seq and Smart-seq2 performs as well as for 10X Genomics with $r^2 = 0.995$ and $r^2 = 0.991$, respectively (Supplementary Table S3). Furthermore, the power calculations are in good agreement with simulation based estimates (Supplementary Fig. S19).

The adapted expression probability models combined with platform-specific sequencing costs, either default (Supplementary Table S5) or user-defined, serve as input to budget optimization. Analogous to (Fig. 5), we evaluated the evolution of parameters for simulated priors and observed priors in Drop-seq and Smart-

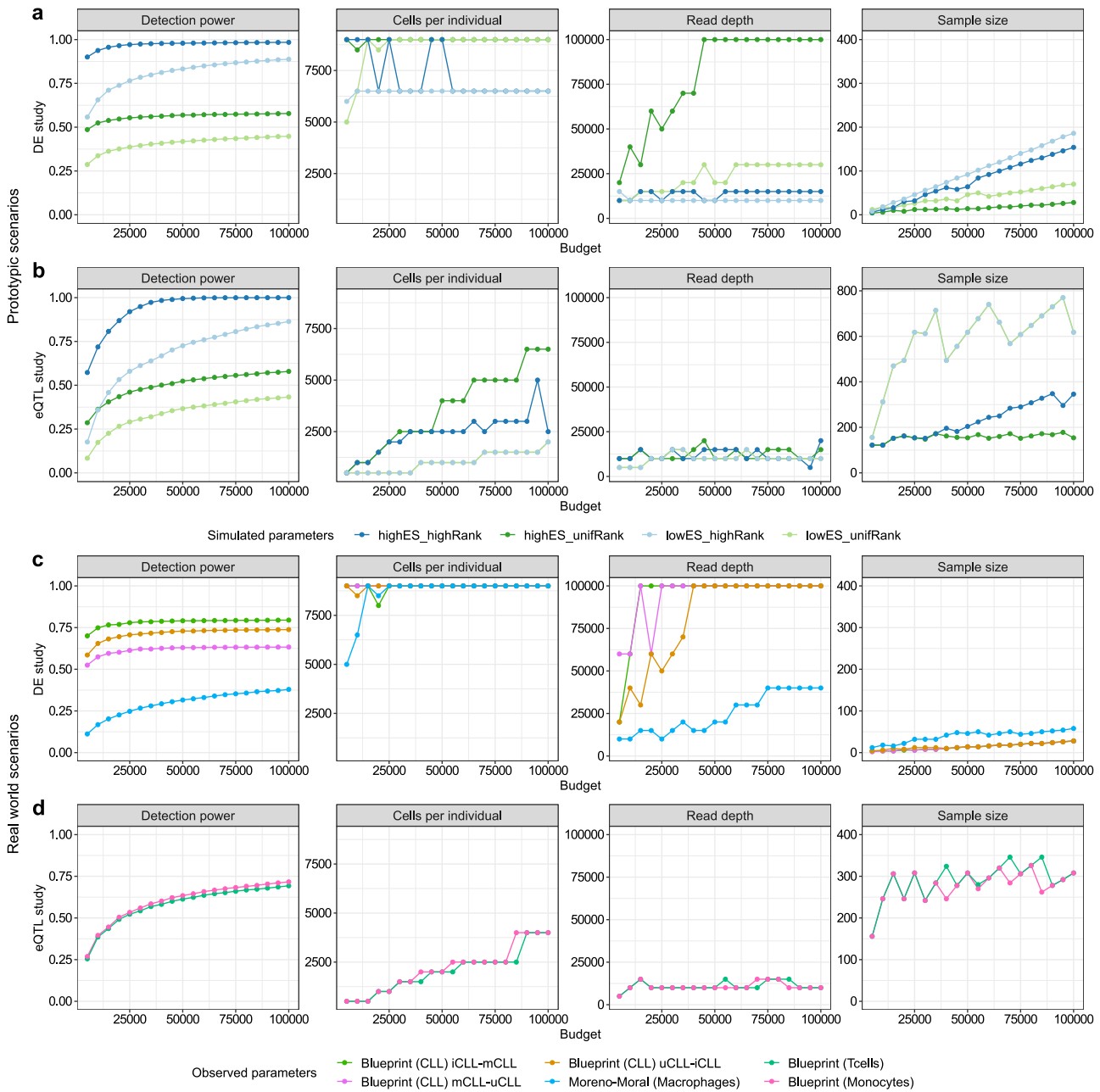

**Fig. 5 Optimal parameters for varying budgets and 10X Genomics data.** The maximal reachable detection power (column 1) and the corresponding optimal parameter combinations (columns 2–4) change depending on the given experimental budget (x-axis). The coloured lines indicate different effect sizes and gene expression rank distributions. Different simulated effect sizes and rank distributions for DEG studies (**a**) and eQTL studies (**b**) with models fitted on 10X PBMC data. highES = high effect sizes, lowES = low effect sizes, highRank = high expression ranks and unifRank = uniformly distributed expression ranks (always relative to effect sizes observed in published studies). Effect sizes and rank distributions observed in cell type sorted bulk RNA-seq DEG studies (**c**) and eQTL studies (**d**) with model fits analogously to (**a, b**). Expression thresholds were chosen as for Fig. 4.

seq2 (Supplementary Fig. S20). For the Smart-seq2 pancreas study, the overall observed power is lower. In contrast to 10X Genomics and Drop-seq, the optimal number of reads per cell is much higher and the number of cells per individual and sample size is only increased at higher budgets for both the prototypic and data driven priors. In general, we observe that Smart-seq2 experiments are not less powerful per se, but the significantly higher cost in the multi-sample setting leads to less powerful designs when restricting the budget. This allows only to measure much fewer cells, even though a higher number of samples and cells would be beneficial. For the Drop-seq lung data we observe

similar trends as for the 10X PBMC data set, with the number of cells per individual being the major determinant of power.

## Discussion

We have introduced scPower, a method for experimental design and power analysis for interindividual differential gene expression and eQTL analysis with cell type resolution. Our model generalizes across different tissues and scRNAseq technologies and provides the means to easily design experiments that maximize the number of biological discoveries.

Previous experimental design methods for multi-sample scRNA-seq[43] are based on simulations. These simulations allow for assessing complex single cell multi-sample data, including scenarios of cell to cell heterogeneities other than differential gene expression. However, analytical models, such as our framework, are by orders of magnitude faster than comparable simulation-based tools. This transformatively enables the evaluation of many experimental design options in a short time and thus to identify optimal experimental parameters for a limited budget. In addition, analytical models require only a small amount of memory independent of the assessed experimental parameters, while simulation of data sets with larger sample sizes lead to increasing memory usage. A sample size of 20 with 3000 cells per sample required already between 35 GB (muscat) and 48GB (powsimR) in our evaluation. Therefore, larger data sets with hundreds of samples, as required for eQTL studies, will be very difficult to simulate.

A first analytic investigation of power optimization in single cell eQTL studies[50] has been done, but suffered from several limitations. First, it was based solely on the effective sample size, ignoring actual effect sizes and expression strength of eQTL genes. Second, it provided no generalizable tool. Third, it is limited to eQTL analysis and does not cover DE studies. In contrast, our approach provides gene level and overall power estimates based on prior data and we provide a generalizable tool for analytic power analysis of single cell DE and eQTL studies. This enables the user the evaluation of his target experiment in order to identify the use-case specific optimal parameter combination. The method is implemented in an R package with a user-friendly graphical user interface and is freely available on github. In addition, the graphical interface of our model is also available over this website http://scpower.helmholtz-muenchen.de/.

We identify the optimal experimental parameters based on expression priors from single cell atlases of three different tissues and cell type specific effect size priors from bulk DEGs and eQTLs. We show that the number of cells is not only crucial for the power to detect rare cell types[37,38] but also for the power to find DE/eQTL genes by increasing the sensitivity of gene expression detection. In line with Mandric et al.[50], our analyses suggest that aggregating shallowly sequenced transcriptomes of a large number of cells of the same cell type is a more cost efficient way than increasing read depth to increase the sensitivity for individual level gene expression analysis. Most likely, multiple independent library preparations in individual cells lead to an improved sampling of the transcriptome as compared to fewer independent libraries sequenced more deeply, an effect that has previously been analyzed in the context of variant detection[79].

Specifically, we found the optimal read depths to be ~10,000 in most evaluations, which is relatively low compared to previous recommendations[34–36,80,81]. However, in a systematic analysis of spike-in expression it has been shown that the accuracy of the measurements is not strongly dependent on the sequencing depth and consistently high for a read depth of 10,000 reads per cell[35]. Hence, we expect to accurately quantify the gene expression levels with the optimized experimental design.

On top of the DE/eQTL power, the number of cells and sequencing depth also determine accurate cell type annotation[82]. Shallow sequencing of more cells has been recommended for extracting the gene expression programs required for annotations, because it has achieved equal accuracy as deeper sequencing of fewer cells[50,82]. These recommendations match our optimal determined parameters. To ensure sufficient power for cell type annotation, our framework scPower can be combined with specific power analysis tools for cell type annotation[38,82].

The optimal sample size is mostly dependent on the effect size, with low effect sizes requiring large sample sizes and consequently optimal setting with high sample size typically lead to low sequencing depth and relatively low number of cells.

In general, priors affect the optimal design and should therefore be selected carefully. In the optimal case, priors are known from well matched pilot experiments or knowledge from the literature. Of note, our data driven priors only allow for reliably assessing the overall power in sample sizes that are smaller or roughly equal to the sample size of the pilot data sets from which the effect sizes were estimated. Consequently, a larger sample size will identify new significant DEGs with lower effect sizes, which were not identified in the smaller pilot study and thus not included in the computation of the overall detection power.

In the absence of well-matched pilot experiments, it is nevertheless important to make assumptions explicit by either selecting a prior based on a similar biological phenomenon or by choosing a prototypic case. In our study, we have compared the prototypic cases of strong effect sizes and relatively high expression versus intermediate effect sizes and expression levels across the whole range from highly expressed to lowly expressed genes. Both options, processing priors from a selected reference study and simulating proteotypic priors, are possible with scPower and described in the package vignette.

The pseudobulk approach presented here leverages well established power analysis methods based on (generalized) linear models. While the (negative binomial) regression model for pseudobulk is currently the most powerful method for assessing individual level differential expression[43], it requires a discrete cell type definition and our approach is tightly linked to it. Therefore, continuous cell annotations such as pseudo time would need to be discretized before the power analysis.

Our model requires the user to choose between our defaults and custom settings for parameters such as doublet rate and expression threshold. The default for the doublet rate is based on reference values from 10X Genomics and is a lower bound compared to the doublet rates we estimate for our own data and to rates reported by other studies[47,78]. Thus, actual experiments might result in higher doublet rates and lower number of usable cells. The choice of a threshold on the number of reads required for a gene to be called expressed also influences the choice of optimal parameters. In our examples we used a threshold of >10 and >3 reads, however, some eQTL analyses of bulk RNAseq data advocate using >0 reads[70].

Following the independent filtering strategy of DESeq2[5,83], we additionally offer users to find the threshold optimizing the number of discoveries at a given FDR (see package vignette). The identified optimal thresholds are low and increase the number of detectable genes. However, the user needs to be aware that this strategy is likely increasing the number of false positives[18]. For this reason, best practice guidelines for differential gene expression with RNA-seq recommend cutoffs that remove between 19 and 33% of lowly expressed genes, depending on the analysis pipeline[84]. These percentages correspond to 1–10 reads per million sequenced, which translates to 1–5 UMI counts for a median of around 5000 UMI counts per cell in our data set. Our gene expression probability model is cell type specific and has to be fitted based on realistic pilot data. We have shown that our model can be applied to data generated with 10X Genomics, Drop-seq and Smart-seq2 and we are confident that it is applicable to similar technology platforms.

When using our approach, the user should keep in mind that our experimental design recommendations are optimized for differential expression between individuals. Other applications might result in very different optimal experimental designs. For instance, co-expression analysis requires a high number of quantified genes per cell, especially when one is interested in cell type specific co-expression and comparison of such co-expression

relations between individuals. Furthermore, the power to annotate new rare cell types by clustering analysis of scRNA-seq data might have different optimal parameters[38]. Lastly, we did not address the power for the detection of variance QTLs (quantitative trait loci associated with gene expression variance across cells) from scRNAseq data[48] due to the lack of data driven priors for the effect sizes.

The human cell atlas project has made it its goal to build a reference map of healthy human cells by iteratively sampling the cells with increasing resolution[61,85]. This will create high quality priors that will further broaden the applicability of scPower. We are convinced that scPower will provide the foundation for building rational experimental design of interindividual gene expression comparisons with cell type resolution across a wide range of organ systems.

## Methods

**Collection of PBMCs.** Blood was collected from healthy control individuals according to the clinical trial protocol of the Biological Classification of Mental Disorders study (BeCOME; ClinicalTrials.gov TRN: NCT03984084) at the Max Planck Institute of Psychiatry[86]. All individuals gave informed consent. Peripheral blood mononuclear cells (PBMCs) were isolated and cryopreserved in RPMI 1640 medium (Sigma-Aldrich) supplemented with 10% Dimethyl Sulfoxide at a concentration of roughly 1 M cells per ml.

**Ethics approval, consent to participate and consent for publication.** All investigations have been carried out in accordance with the Declaration of Helsinki, including written informed consent of all participants. Study conduct complies with the recommendations by the ethics committee of the Ludwig-Maximilian University, Munich. Applicable national and EU law, in particular the General Data Protection Regulation (GDPR) (Regulation (EU) 2016/679) has been followed.

Permission for using the data has been obtained from the Biobank of Max Planck Institute of Psychiatry. Consent for secondary use of the existing data has been obtained. In compliance with the consent for secondary use, the data generated in this project will be stored and can be used for future research. All data has been pseudonymized. Written informed consent of all participants allows for publication of data in online repositories.

**Single cell RNA-sequencing.** For single-cell experiments, 14 cell vials from different individuals (7 male and 7 female) were snap-thawed in a 37 °C water bath and serially diluted in RPMI 1640 medium (Sigma-Aldrich) supplemented with 10% Foetal Bovine Serum (Sigma-Aldrich) medium. Cells were counted and equal cell numbers per individual were pooled in two pools of 7 individuals each. Cell pools were concentrated and resuspended in PBS supplemented with 0.04 % bovine serum albumin, and loaded separately or as a combined pool with cells of all 14 individuals on the Chromium microfluidic system (10X Genomics) aiming for 8000 or 25,000 cells per run. Single cell libraries were generated using the Chromium Single Cell 3′library and gel bead kit v2 (PN #120237) from 10X Genomics. The cells were sequenced with a targeted depth of ~50,000 reads per cell on the HiSeq4000 (Illumina) with 150 bp paired-end sequencing of read2 (exact numbers for each run in Supplementary Table S2).

**Preprocessing of the single cell RNA-seq data.** We mapped the single cell RNA-seq reads to the hg19 reference genome using CellRanger version 2.0.0 and 2.1.1[87]. Demuxlet version 1.0 was used to identify doublets and to assign cells to the correct donors[47]. In addition, Scrublet version 0.1 was run with a doublet threshold of 0.28 to identify also doublets from cells which originate from the same donor[88]. Afterwards, the derived gene count matrices from CellRanger were loaded into Scanpy version 1.4[89]. Cells identified as doublets or ambivalent by Demuxlet and Scrublet were removed, as well as cells with less than 200 genes or more than 2,500 genes and with more than 10% counts from mitochondrial genes.

**Verification of Demuxlet assignment using sex-specific errors.** We validated the donor assignment and doublet detection of Demuxlet by testing if assigned cells express sex-specific genes correctly. Xist expression was taken as evidence for a female cell, expression of genes on the Y chromosome as evidence for a male cell.

The male-specific error shows the fraction of cells assigned to a male donor among all cells expressing Xist (count > 0). The threshold for the female-specific error was set less strictly, as mismapping of a few reads to the chromosome Y occurs also in female cells. Instead, the female-specific error indicates which fraction of cells is assigned to a female donor among all cells having more reads mapped to chromosome Y than the $q_f$ quantile of all cells, with $q_f$ being the overall fraction of cells assigned to a female donor among all cells. TPM mapped to

chromosome Y is calculated by counting all reads mapped to chromosome Y, excluding reads mapped to the pseudoautosomal regions, times $10^6$ divided by the total number of read counts per cell.

Both error rates are calculated twice, once with all cells and once without doublets from Demuxlet and Scrublet.

**Cell type identification.** We performed the cell type identification according to the Scanpy PBMC tutorial[90]. Genes which occurred in less than 3 cells were removed. Counts were normalized per cell and logarithmized. Afterwards the highly variable genes were identified, the effect of counts and mitochondrial percentage regressed out. We calculated a nearest neighbour graph between the cells, taking the first 40 PCs, and then clustered the cells with a Louvain clustering[91]. Cell types were assigned to the clusters using marker genes (Supplementary Table S6).

**Frequency of the rarest cell type.** The probability to detect at least $n_{c,s}$ cells of a specific cell type $c$ in each individual $s$ depends on the frequency of the cell type $f_c$, the number of cells per individual $n_c$ and the number of individuals $n_s$. For one individual, the minimal number of cells can be modeled using a cumulative negative binomial distribution[37] as $F_{NB}(n_c - n_{c,s}, n_{c,s}, f_c)$ and for all individuals as $F_{NB}(n_c - n_{c,s}, n_{c,s}, f_c)^{n_s}$.

The cell type frequencies were obtained by literature research, the frequencies in PBMC are approximately twice as high as in whole blood[92]. All other parameters can be freely chosen (dependent on the expected study design).

**Influence of read depths.** We used subsampling to estimate the dependence of gene expression probabilities on read depths. The fastq files of all 6 runs were subsampled using fastq-sample from fastq-tools version 0.8[93]. The number of reads was downsampled to 75%, 50% and 25% of the original number of reads. Cell-Ranger was used to generate count matrices from the subsampled reads. Donor, doublet and cell type annotation were always taken from the full runs with all reads.

**Expression probability model.** The gene expression distribution of each cell type was modeled separately because there are deviations in RNA content between different cell types[33]. The UMI counts $x$ per gene across the cells of a cell type are modeled by a negative binomial distribution. We used DESeq[60] to perform the library size normalization as well as the estimation of the negative binomial parameters. The standard library size normalization of DEseq and the variant "poscounts" of DESeq2[5] were both used, depending on the quality of the fit for the specific data set. For the PBMC 10X data set (Supplementary Table S2), the standard normalization was taken, for the Drop-seq lung and the Smart-seq2 pancreas datasets the poscount normalization, which is more suitable for sparse data. Only cell types with at least 50 cells were analyzed to get a robust estimation of the parameters. Negative binomial distributions were fitted separately for each batch to avoid overdispersion by batch effects and the fits combined downstream (see paragraph about gamma mixture distribution).

The negative binomial distribution is defined by the probability of success $p$ and the number of successes $r$:

$$f_{NB}(x, r, p) = NB(x, r, p) = \binom{x + r - 1}{x} \cdot (1 - p)^r \cdot p^x \quad (3)$$

DESeq uses a parametrization based on mean $\mu = \frac{p \cdot r}{1 - p}$ and dispersion parameter $\phi = \frac{1}{r}$.

We formulated the definition of an expressed gene in a flexible way so that users can adapt the thresholds. The definition is based on the pseudobulk approach where the counts $x_{i,j}$ are summed up per gene $i$ for all cells $j$ part of cell type $c$ and donor $s$ to a three dimensional matrix $y_{i,c,s} = \sum_{j \in C \wedge j \in S} x_{i,j}$ with $C$ the set of all cells part of cell type $c$ and $S$ the set of all cells part of donor $s$.

In general, a gene $i$ is called expressed in a cell type $c$ if the sum of counts $y_{i,c,s}$ over all cells of the cell type within an individual $s$ is greater than $n$ in more than $k$ percent of the individuals.

We assume a negative binomial distribution ($f_{NB}(x_{i,j}, \mu_{i,c}, \phi_{i,c})$) for the counts $x_{i,j}$ of each gene $i$ in each cell type $c$ with $\mu_{i,c}$ and $\phi_{i,c}$. The sum of gene counts $y_{i,c,s}$ follows a negative binomial distribution where the parameters are altered by the number of cells per cell type and donor $n_{c,s} = |\{j \in C \wedge j \in S\}|$ to $\mu'_{i,c,s} = n_{c,s} \cdot \mu_{i,c}$ and $\phi'_{i,c,s} = \frac{\phi_{i,c}}{n_{c,s}}$. The probability that the sum of counts $y$ is greater than $n$ is

$$p_{i,s} = P(y_{i,c,s} > n) = 1 - F_{NB}(n, \mu'_{i,c,s}, \phi'_{i,c,s'}) \quad (4)$$

with $F_{NB}$ as the cumulative negative binomial distribution.

To define a gene as expressed, we require that it can be found in a certain fraction of more than $k$ percent in all $n_s$ individuals. The expression probability of a gene $i$ is obtained from a cumulative binomial distribution $F_{Bin}$ as

$$P(i \in E) = 1 - F_{Bin}(k \cdot n_s, n_s, p_{i,s}) \quad (5)$$

So in total, the expected value of the number of expressed genes ($E$) can be defined as

$$\mathbb{E}(E) = \sum_{gene\,i} P(i \in E) \qquad (6)$$

To generalize the expression probability model also for unseen data sets, the distribution of the mean values $\mu_{i,c}$ over all genes in a cell type $c$ is modelled as a mixture distribution with three components, a zero component $Z(x)$ and two left-censored gamma distributions $\Gamma(x, r, s)$:

$$f_{\mu_i}(x) = p_1 Z(x) + p_2 \Gamma(x, r_1, s_1) + p_3 \Gamma(x, r_2, s_2) \qquad (7)$$

The model is an adaptation of the distribution used in the single cell simulation tool Splatter[94]. The largest part of the mean values can be fitted with one gamma distribution, a small fraction with high expressed gene outlier with the second gamma distribution. The genes with zero mean values originate from two sources. Either, the gene is not expressed or the expression level is too low to be captured in the setting. The lower bound for the expression level at which both Gamma distributions are censored depends on the number of cells $j$ measured for this cell type $n_c = |\{j \in C\}|$. The smallest expression level to be captured is $\frac{1}{n_c}$.

The density of the gamma distribution is parametrized by rate $r$ and shape $s$:

$$\Gamma(x, r, s) = \frac{s^r x^{r-1} e^{-sx}}{(r-1)!} \qquad (8)$$

For modeling of the gamma parameters, also the parameterization by mean $\mu = \frac{s}{r}$ and standard deviation $\sigma = \sqrt{\frac{s}{r^2}}$ is used.

The relationship between the mean UMI counts per cell and the gamma parameters (mean and standard deviation of the two gamma distributions) is linear and $\beta$ values are estimated by linear regression, fitted over the gamma distribution for each run and all subsampled runs. The mixture proportion of the zero component $p_1$ is linearly decreasing with the mean UMI counts, also estimated by linear regression. The lower bound of $p_1$ is set to a small positive number: 0.01. In contrast, the mixture proportion of the second gamma component $p_3$ is modelled as a constant, independent of the mean UMI counts. We set it to the median value of all fits per cell type. The mixture proportion of the first gamma component is $p_2 = 1 - p_1 - p_2$ and is linearly increasing with increasing mean UMI counts.

The number of transcriptome mapped reads is linearly related to the logarithm of the mean UMI counts per cell, with an increasing read depth leading to a saturation of UMIs. 10X Genomics describes this also with the sequence saturation parameter. The exact logarithmic saturation curve depends on multiple biological and technical factors, therefore it needs to be fitted for each experiment individually. However, scPower provides example fits from the different scenarios observed in our analysis.

The dispersion parameter is estimated dependent on the mean value using the dispersion function fitted by DESeq. The parameters of the mean-dispersion curve showed no correlation with the mean UMI counts, therefore the mean of the parameters of the dispersion function across all runs and subsampled runs were taken, resulting in one mean-dispersion function per cell type.

**Expression cutoffs and threshold criteria.** The selection of expression cutoffs both on the individual level and the population level depends on the users and their research question, balancing the increase in power by more lenient cutoffs and the potential higher false positive rates associated with it. We applied different UMI count cutoffs for the individual level to prove the flexibility of our tool.

For the population level, we followed in most of our analyses the recommendation of edgeR[75] that the expression cutoff should correspond to the percentage of samples in the smaller group. In the DE case, this results in a cutoff of 50% as we focus on studies with balanced design. In the eQTL case, the definition of groups depends on the genotype and is therefore not directly chosen by the user. We decided to select the cutoff based on the minor allele frequency $f_A$, so that at least in heterozygotes the gene should be expressed. The fraction of heterozygotes $f_{AB}$ is thereby calculated dependent on the minor allele frequency as:

$$f_{AB} = 2 * f_A * (1 - f_A) \qquad (9)$$

For example, assuming a minor allele frequency of at least 0.05 would result in a population cutoff of 0.095.

Furthermore, our R package provides alternative threshold criteria. On the population level, instead of a percentage threshold for the number of individuals, an absolute threshold can be chosen. On the individual level, instead of an absolute count threshold in the pseudobulk, a gene can be defined as expressed if it is expressed in a certain number of cells with count larger than 0. Both alternative criteria are based on the same model as explained above in the previous section.

If the users want to choose a threshold that maximizes the power, our package provides an optimization function for that.

**Power analysis for differential expression.** The power to detect differential expression, also denoted as the probability $P(i \in S)$ that gene $i$ is in the set of significant differentially expressed genes $S$, is calculated analytically for the negative binomial model[64]. An implementation of the method can be found in the R package MKmisc. Parameters are sample size, fold change, significance threshold, the mean of the control group, the dispersion parameter (assuming the same

dispersion for both groups) and the sample size ratio between both groups. We focus in our analyses on balanced comparisons with the same number of samples in both groups, represented by a sample size ratio of 1. Zhu et al. implemented three different methods to estimate the dispersion parameter, we chose method 3 for the power calculation, which was shown to be more accurate in simulation studies in the paper. More complex experimental designs can be addressed using the method of[95].

**Power analysis for expression quantitative trait loci.** Additionally to the DE analyses, the use of scRNA-seq for the detection of expression quantitative trait loci (eQTLs) was evaluated. We distinguish for the eQTL power between genes with high and with low expression levels, where the mean is used to parameterize a simulation. Therefore, the eQTL power is a function of the mean expression level.

For genes with high expression level, the power to detect an eQTL is calculated analytically using an F-test and depends on the sample size $n_s$, the coefficient of determination $R^2$ of the locus and the chosen significance threshold $\alpha$. $R^2$ is calculated for the pilot studies from the regression parameter $\beta$, its standard error $se(\beta)$ and the sample size $N$ of the pilot study:

$$t = \frac{\beta}{se(\beta)} \qquad (10)$$

$$R^2 = \frac{t^2}{N - 2 + t^2} \qquad (11)$$

The implementation pwr.f2.test of the R package pwr is used for the F-test[20]. The degrees of freedom of the numerator are 1 and of the denominator are $n_s - 2$, the effect size is $\frac{R^2}{1-R^2}$.

This power calculation assumes that the residuals are i.i.d. normally distributed. For large count values, it has been shown that normalized log transformed counts have a constant variance independent of the mean value and can be analyzed with linear models[45]. However, for genes with small mean values, i.e., only very few non-zero counts, this normalization might not be effective and the power is overestimated by the analytical power calculation based on the F-test. We performed a simulation study to assess the effect of the mean values on the eQTL power.

To account for the discrete nature of the counts we adopted a simulation scheme similar to a negative binomial regression model and analyzed the log transformed counts using linear models[45]. As for the analytical power calculation, the effect size is given by the coefficient of determination $R^2$ of the locus. To determine the simulation based power for sample size $n_s$, significance threshold $\alpha$ and mean count $\mu_c$ of the allele with lower expression, the following steps are repeated B = 100 times:

1. Simulate genotypes. To also account for the discrete nature of the genotypes, we first draw allele frequency $f_a$ from a uniform distribution between 0.1 and 0.9. A random genotype vector $g$ with $g_i \in \{0, 1, 2\}$ of length $n_s$ is generated with the expected number of each genotype $(f_a^2, 2f_a(1 - f_a), (1 - f_a)^2)$ according to Hardy Weinberg equilibrium.

2. Simulate read counts. Using the allele frequency, the beta value $\beta$ and the standard deviation of the residuals $\hat{\sigma}$ is calculated:

$$\beta = \sqrt{\frac{R^2}{2 * f_a * (1 - f_a)}} \qquad (12)$$

$$\hat{\sigma} = \sqrt{1 - R^2} \qquad (13)$$

The associated gene expression count vector $x$ is sampled from a negative binomial distribution parameterized for each genotype $g_i$ with mean $\mu_i = e^{\log(\mu_c) + \beta * g_i}$ and dispersion $\phi_i$. In the following, we work with log transformed counts (plus one pseudo count). To match with the effect size $R^2$, the dispersion parameter $\phi_i$ is chosen, such that the variance of the log transformed counts is $\hat{\sigma}$. Since the Taylor approximation of the dispersion parameter[45] was not accurate enough, we used instead a numerical optimization. This numerical optimization is precalculated for a range of parameter combinations to speed up calculation for the user.

Using the linear regression $\log(x_i + 1) \sim g_i$, the $p$ value $P_i$ for $H_o : \beta = 0$ is determined.

Finally, the simulation based power is estimated as $\sum\limits_{i=1}^{B} P_i < \alpha$

The power of the simulation was compared with the analytic power calculated by scPower to assess at which value of the mean $\mu_c$ the analytic power starts to overestimate the simulation based empirical power (see Supplementary Fig. S12) for Bonferroni adjusted significance thresholds used in eQTL analyses. We choose a cut-off of mean count < 5 and estimate the power for genes with smaller mean values based on simulation instead of the F-test to increase accuracy for small count values.

**Overall detection power.** The overall detection power for DEGs/eQTLs is the product of the expression probability and power to detect DEGs/eQTLs, as both probabilities are conditionally independent given the expression mean of the gene.

Expression probabilities were determined based on the gene expression rank in the observed (pilot) data. The number of considered genes $G$ was set to 21,000, the number of genes used for fitting of the curves. Ranks $i$ were transformed to the quantiles $\frac{i}{G}$ of the gamma mixture model parameterized by the mean UMI counts to obtain the mean $\mu_c$ of the negative binomial model, which is in turn used to compute the expression probability.

To quantify the overall power of an experimental setup, we compute the expected fraction of detected DEG/eQTL genes with prior expression levels and effect sizes derived from the pilot data. We obtain gene expression ranks of DEGs/eQTLs and their corresponding fold changes to compute overall detection power for each gene. The overall power of the experimental setup is then the average detection power over all prior DEG/eQTL genes.

DE/eQTL power is computed using a significance threshold $\alpha$ corrected for multiple testing, controlling either the family-wise error rate (FWER) or the false discovery rate (FDR). We used FDR adjustment for the DE power and followed the approach of the GTEx consortium[63] based on FWER adjustment for the eQTL power. However, our framework allows for any combination of power analysis and multiple testing method. For all analyses shown, adjusted $\alpha$ was set to 0.05.

The family-wise error rate is defined as the probability of at least one false positive among all tests. Each expressed gene is tested once in the DE analysis, therefore, the adjustment for the family-wise error rate is done by correcting the threshold to $\frac{\alpha}{(E)}$ for $(E)$ expected expressed genes. For eQTLs we followed the approach of the GTEx consortium[63], which assumes that for each gene on average 10 independent (uncorrelated) SNPs are tested in a genome-wide cis eQTL analysis. Thus, the adjusted $p$ value threshold is set at $\frac{0.05}{(E)*10}$.

In our tool, the number of independent SNPs can be flexibly chosen, for example if the user wants to perform a cis and trans eQTL analysis, he can define a higher number of independent SNPs.

Alternatively, for DE analysis the significance threshold can be adjusted for the false discovery rate using the method of Jung[22]. In contrast to the Bonferroni correction, which depends only on the number of tests, the FDR correction depends on the $p$ value distribution of all genes. As our analytic method outputs the power without computing the $p$ values, we can not apply the FDR correction directly and use the method of Jung.

The goal of the approach is to identify the raw $p$ value $\alpha'$ corresponding to chosen FDR corrected threshold $\alpha = FDR(\alpha')$.

The FDR is the fraction of false positives among all rejected null hypotheses (predicted positives), which includes both the false positives and the true positives. Based on the probability integral transform, the distribution of $p$ values for the $m_o$ true null hypotheses is uniform. Therefore, we expect $m_o * \alpha'$ false positives at a raw $p$ value significance threshold of $\alpha'$. Here $m_o = (E) - (E_{DEG/eQTL})$ is the number of expected expressed genes without the expected expressed DEGs/eQTLs.

The expected number of true positives is directly derived from the power we reach for $\alpha'$. Summing up the gene-wise power (at $\alpha'$) yields the expected number of significant DEGs/eQTLs $r_1(\alpha')$.

Using numerical optimization of the complete formula

$$FDR(\alpha') = \frac{m_0 * \alpha'}{m_0 * \alpha' + r_1(\alpha')} \qquad (14)$$

with respect to the unknown parameter $\alpha'$ we identify the raw $p$ value threshold $\alpha'$ corresponding to the FDR threshold of $\alpha$.

**Pilot data sets**. Realistic DE and eQTL priors, i.e., effect sizes and expression ranks, were taken from sorted bulk RNA-seq studies of matching tissues (PBMCs, lung and pancreas). For all studies, the significance cut-off of the DE and eQTL genes was set to FDR <0.05 and the expression levels of the genes were taken from FPKM normalized values. When published, we took directly the effect sizes, otherwise we recalculated the DE analysis with DEseq2.

Differential gene expression: To get realistic estimates for effect sizes (fold changes), data sets from FACS sorted bulk RNA-seq studies were taken[53,54]. The data sets were used to rank the expression level of the DEGs among all other genes using the FPKM values. The cell types used in the studies were matched to our annotated cell types in PBMCs for the expression profiles. The expression profile of CD14+ Monocytes was used for the study of Macrophages, the profile of CD4+ T cells for the CLL study.

Lung cell type specific priors were obtained from a DE study of freshly isolated airway epithelial cells of asthma patients and healthy controls[55]. As no effect sizes were reported, the analysis was redone with the given count matrix from GEO (accession number GSE85567) using DEseq2.

A DE study analyzing age-dependent gene regulation in human pancreas[56] was used to get pancreas cell type specific priors. We obtained expression ranks and gene length, which is needed for proper normalization of Smart-seq2 expression values.

eQTLs: We used eQTL effect and sample sizes from the Blueprint study on bulk RNA-seq of FACS sorted Monocytes and T cells[57]. Neutrophils were excluded as they are not PBMCs. We took the most significant eQTL for each gene, using a significance cutoff of $10^{-6}$. We compared the FPKM normalized expression levels of the eQTL genes among all other genes to get the expression rank for each eQTL

gene. Effect sizes were derived from the slope parameters of the linear regression against genotype dosage, its standard error and the sample size of the study.

**Comparison with simulation-based power analysis tools**. To validate our model, we compared the DE power estimations of our framework with two simulation-based tools, called powsimR and muscat[40,43]. For both tools, a few changes needed to be implemented to compare the output exactly with our approach. powsimR is not designed for multi-sample comparison and for both methods the option to apply a vector of log-fold changes with matching expression ranks was not available. A detailed explanation of both methods and applied changes can be found below.

The simulation-based methods perform random sampling of their count matrices and therefore the simulation was repeated 25 times for each parameter combination to generate stable results. Both tools allow the power estimation for different DE methods. We evaluated powsimR in combination with edgeR-LRT, DESeq and limma-voom, together with median-ratio normalization of DESeq ('MR'), and muscat in combination with edgeR, DESeq2, limma-voom and limma-trend. No imputation or filtering was applied for any of the methods. In the comparisons with our model scPower, the expression probability parameters of scPower were set to minCounts >0 in at least one individual to match the detected genes of powsimR and muscat. Exemplarily, the CD4 T cells of our PBMC data set were used for fitting the simulation models of powsimR and muscat. We evaluated all DE methods for 4, 8 and 16 samples in combination with 200, 1000 and 3000 cells per person. Additionally, we performed a comparison for a large range of parameter combinations of powsimR with edgeR-LRT and muscat with edgeR, testing all combinations as evaluated in (Fig. 3a).

In the following, it is important to distinguish the training data set, which is used for model fitting of powsimR/muscat and restricts the number of simulated genes, and the simulated data set which is sampled from the trained model.

The three main components of our statistical framework were evaluated in the comparison, the expression probability (by comparing the number of expressed genes), the power (here according to the definition of powsimR, i.e., the power of all genes expressed in the simulated data) and the overall detection power.

1. Expressed genes: The expected number of expressed genes for scPower is compared with the number of expressed genes in powsimR and muscat, which are all genes with at least one count in the simulated matrix. An important limitation of the simulation based frameworks is here that the number of expressed genes in the simulation tools can never be larger than the number of expressed genes in the training data set, while scPower can also approximate expression of unseen genes with smaller mean values and so estimate more expressed genes than seen in the pilot data.

2. DE power: The reported power of powsimR includes only genes, which are expressed in the simulated data set (count > 0). The same value can also be calculated for muscat. To make the DE power of our framework comparable, the mean power for all expressed DE genes was calculated. An expressed DE gene for scPower is defined by its expression rank, which needs to be smaller than the expected number of expressed genes.

3. Overall power: powsimR does not return directly an overall power, which we define as the power over all simulated DE genes (including genes simulated with count > 0 and count = 0). However, the overall detection power of powsimR can simply be calculated by dividing the number of true positives of powsimR by the number of all simulated DE genes. The same was done for muscat.

powsimR: uses training data to fit the parameters of the expression distributions for each cell type and gene. Using these parameters, it is randomly generating count matrices for two groups of cells introducing differential gene expression between these two groups for a prespecified number of DE genes. These DE genes are randomly selected and the means of their distributions shifted by a given effect size. In the next step the simulated data is analyzed with different methods and results are compared to the simulated group truth to determine the power.

Adaptations of powsimR are required to simulate a multisample setting and thus make it comparable to scPower: We added an additional step that generates a pseudobulk count matrix for multi-sample comparison. For this, we included an additional parameter for the sample size $n_s$, with samples distributed equally across both groups ($n_s/2$ samples per group). Thus, individual level effect sizes are identical to the cell level effect sizes, as more complex differential distributions are not implemented in powsimR[96]. After simulation of the new count matrix $C$ with dimensions $n_C$ (number of cells) times $n_G$ (number of genes) in powsimR, we changed the algorithm to equally distribute the simulated cells between the samples ($n_C/n_s$ cells for each sample), while preserving the group structure. Summing up the counts for each sample generates a pseudobulk matrix with dimensions $n_s$ times $n_G$, which can be processed exactly the same way as a single cell matrix in the following steps in powsimR. Furthermore, instead of randomly sampling the position of the DE genes with powsimR, we assigned DE genes based on their expression ranks in the bulk studies, as in scPower.

muscat: was specifically implemented for multi-sample comparisons, in contrast to powsimR. It fits one negative binomial distribution separately for each sample and subpopulation in the training data set. The subpopulation definition is hereby

equivalent to our cell type definition. We noticed that fitting each sample separately decreases the number of expressed genes quite drastically, if not a sufficient number of cells are available for each sample. Because again only as many genes can be sampled as are detected in the training data set. To get a robust fit of the negative binomial distribution with our training data set, we therefore decided to fit the negative binomial distribution for all samples together, for a very large training data set this is probably not necessary.

Another difference to powsimR is that muscat provides different scenarios for simulating differential expression besides the shift of the mean expression (called DE in muscat). Additionally, they simulate genes with different proportions of low and high expression-state (DP), differential modality (DM) or changes in both proportions and modality (DB). For comparison with powsimR and scPower, we focus on the DE scenario.

Similar to powsimR, we also incorporated here the option to assign genes of a specific expression rank a specific log fold change to simulate the same DE genes as in scPower.

We applied the simulation-based methods to further validate the scPower estimations in regards to two different real life scenarios that can affect the power: batch effects and differences of cell proportions between the groups. The introduction of batch effects was already implemented in powsimR. Of note, we slightly adapted the code of powsimR again to the multi-sample setting by assigning all cells from one individual to the same batch. We separated the individuals into two different batches with 50% cases and 50% controls in each batch, which represent a setup based on a non-confounded experimental design. 20% of the genes were randomly sampled to show batch log fold changes with values between 2 and 6. We ran the downstream powsimR power analysis with edgeR once without accounting for batch effects and once with adjusting for the batch effects using a model that includes a batch covariate. This was repeated for different batch effects and experimental parameter combinations and each time the results were compared with the scPower estimation.

The second real life scenario simulating differences of cell proportions between the groups is readily implemented in muscat. The cell proportion parameter represents the fraction of all measured cells that belong to group 1, i.e., a fraction of 0.3 means that 30% of measured cells belong to group 1 and 70% to group 2. We evaluated cell proportions between 0.1 and 0.5 in combination with different experimental parameter combinations and compared the results to scPower. For the scPower estimation, we calculated two versions, once the default approach assuming balanced distribution of cells between the groups and once a conservative approach, also assuming the balanced distribution, but reducing the cell frequency $f_c$ by the cell proportion parameter $p_c$ so that the number of cells per cell type entering the model matches the cell frequency in the lower group $2 * f_c * p_c$.

As no simulation-based power analysis for eQTLs exists (and also no other method), we benchmarked the eQTL power with our own simulation tool (described in the methods section Power analysis for expression quantitative trait loci). Our simulation method uses our expression probability model to estimate the mean parameter, therefore only the power itself is compared (not the expression probability and overall power). We tested again 25 rounds of simulation for all parameter combinations depicted in Fig. 3b.

**Cost calculation and parameter optimization for a given budget.** The overall experimental cost $C_t$ for a 10X Genomics experiment is the sum of the library preparation cost and the sequencing cost. It can be calculated dependent on the three cost determining parameters sample size $n_s$, number of cells per sample $n_c$ and the read depth $r$. The library preparation cost is determined by the number of 10X kits, depending on how many samples are loaded per lane $n_{s,l}$ and the cost of one kit $C_k$. The cost of a flow cell $C_f$ and the number of reads per flow cell $r_f$ determine the sequencing cost.

$$C_t = ceiling\left(\frac{n_s}{6 * n_{s,l}}\right) * C_k + ceiling\left(\frac{n_s * n_c * r}{r_f}\right) * C_f \qquad (15)$$

We optimized the three cost parameters for a fixed budget to maximize the detection power. A grid of values for number of cells per individual and for the read depth was tested, while the sample size is uniquely determined given the other two parameters and the fixed total costs. As an approximation of the sample size, the ceiling functions from the cost formula were removed.

$$n_s = floor\left(C_t \middle/ \left(\frac{C_k}{6 * n_{s,l}} + \frac{n_c * r * C_f}{r_f}\right)\right) \qquad (16)$$

The same approach can also be used with a grid of sample size and cells per sample or read depth. In general, two parameters need to be chosen and the third parameter is uniquely determined given the other two and the fixed experimental cost.

Given the three cost parameters, the detection power for a specific cell type and a specific DE or eQTL study can be estimated. However, we also have to account for the appearance of doublets during the experiment. The fraction of doublets depends on the number of cells loaded on the lane. Following the approach of[37], we model the doublet rate $d$ linear dependent on the number of recovered cells, using

the values from the 10X User guide of V3[77]. A factor of $7.67 * 10^{-6}$ was estimated, so that $d = 7.67 * 10^{-6} * n_c * n_{s,l}$.

The number of usable cells per individual used for the calculation of detection power is then $n_u = (1 - d) * n_c$. We assume that nearly all doublets are detectable using Demuxlet and Scrublet and discarded during the preprocessing of the data set. The expected number of cells for the target cell type with a frequency of $f_c$ will be $f_c * (1 - d) * n_c$.

A second effect of doublets is that the read distribution is shifted, as doublets contain more reads than singlets. Again following the approach of[37], we assume that doublets contain 80% more reads than singlets. In the following, the ratio of reads in doublets compared to reads in singlets is called doublet factor $f_d$, a factor of 1.8 is assumed in the calculations in this manuscript. Therefore, depending on the number of doublets, the read depth of the singlets will be slightly lower than the target read depth.

$$r_s = \frac{r * n_c}{n_u + f_d * (n_c - n_u)} \qquad (17)$$

In addition, the mapping efficiency is taken into account. Assuming a mapping efficiency of 80%, $r_m = 0.8 * r_s$ mapped read depth remains. In the power calculation, the number of usable cells per cell type will be used instead of the number of cells and the mapped read depth instead of the target read depth.

Instead of defining the number of samples per lane directly, usually the number of cells loaded per lane $n_{c,l}$ is defined. So, the doublet rate per lane can be directly restricted. We use in our analyses $n_{c,l} = 20,000$, which leads to a doublet rate of at most 15.4%. The number of individuals per lane can be derived directly as $n_{s,l} = floor(n_{c,l}/n_c)$.

**Simulation of effect sizes and gene rank distributions.** Model priors, i.e., effect sizes and gene rank distributions, were derived from FACS sorted bulk RNA-seq to get realistic assumptions. In addition, we simulated different extreme prior distributions to evaluate their influence on the optimal experimental parameters. The log fold changes for the DE studies were modeled as normally distributed. High effect size distributions were simulated with a mean of 2 and a standard deviation of 1, low effect sizes distributions with a mean of 0.5 and standard deviation of 1.

Effect sizes ($R^2$ values) for the eQTL studies were obtained by sampling normally distributed Z scores and applying the inverse Fisher Z Transformation. Because very small values are not observed due to the significance threshold, the normal distribution is truncated to retain values above the mean. High effect sizes were simulated with a mean of 0.5 and standard deviation of 0.2, low effect sizes with a mean of 0.2 and a standard deviation of 0.2. A similar standard deviation was also observed in the pilot data.

250 DEGs were simulated and 2000 eQTL genes. The ranks were uniformly distributed, either over the first 10,000 genes or the first 20,000 genes. This leads to four simulation scenarios for each, high and low effect sizes (ES) and high or uniformly distributed expression ranks, called in the studies highES_highRank, lowES_highRank, highES_unifRank and lowES_unifRank.

**Evaluation of Drop-seq and Smart-seq2 data.** We validated our expression probability model for other tissues and single cell RNA-seq technologies. Two data sets of the human cell atlas were used for that, a Drop-seq data set measured in lung tissue[52] and a Smart-seq2 data set measured in pancreas tissue[51].

The Drop-seq technology is also a droplet-based technique, similar to 10X Genomics. The same model can be used, only adapting the doublet and cost parameter. However, as there was no data available to model the linear increase of the doublet rate during overloading correctly, the doublet rate was modeled instead as a constant factor and the library preparation costs were estimated per cell. scPower provides models for both cases and with the necessary prior data, users can also model the overloading for Drop-seq.

Smart-seq2 is a plate-based technique, which produces full length transcripts and read counts instead of UMI counts. To compensate the gene length bias in the counts, the definition of an expressed gene was adapted to at least $n$ counts per kilobase of transcript, resulting in a gene specific threshold of $\frac{n*1000}{l_i}$ with $l_i$ as gene length for gene $i$. The gamma mixed distribution of the mean gene expression levels is modelled using length normalized counts, but the gene length is required as a prior for the dispersion estimation and the power calculation, as DEseq uses counts, which are not normalized for gene length. These priors can be obtained together with the effect sizes and the expression ranks from the pilot bulk studies. In the simulation of non-DE genes, an average mean length of 5000 bp is assumed. The linear relationship of the parameters of the mixture of gamma distributions is modeled directly based on the mean number of reads per cell. Doublets also appear in Smart-seq2, but as a constant factor, not increasing with a higher number of cells per individual. We observed for the parameter of the DEseq dispersion model a linear relationship with the read depth, which was not visible for Drop-seq and 10X Genomics. So, instead of taking the mean value per cell type, a linear fit is modeled for Smart-seq2.

For both data sets, the cell type frequencies varied greatly among individuals, therefore an estimation of expressed genes in a certain fraction of individuals could not be validated, as this requires similar cell type frequencies for each donor.

Instead, the expressed genes were estimated to be above a certain count threshold in all cells of a cell type, independent of the individual.

Both data sets were subsampled to investigate the effect of the read depth. The Drop-seq reads are subsampled using fastq-tools version 0.8[93] and the subsampled UMI count matrix was generated following the pipeline previously described in[97]. The Smart-seq2 read matrix was subsampled directly using the function *downsampleMatrix* of the package *DropletUtils*[98].

We compared the budget restricted power to our PBMC 10X Genomics results, using the same simulated effect sizes and distribution ranks as well as matched observed priors from FACS sorted bulk studies.

**Reporting summary**. Further information on research design is available in the Nature Research Reporting Summary linked to this article.

## Data availability

The single cell PBMC data set generated and analysed during the current study is available on Gene Expression Omnibus (GEO) with accession number GSE185714. The other single cell test data sets are available on GEO with accession numbers GSE96583, GSE130148 and GSE81547. The effect sizes for the eQTL and DE power were taken from published studies, accessible in the supplement of Chen et al. 2016 (https://doi.org/10.1016/j.cell.2016.10.026), Rendeiro et al. 2016 (https://doi.org/10.1038/ncomms11938), Moreno-Moral et al. 2018 (https://doi.org/10.1136/annrheumdis-2017-212454) and Arda et al, 2016 (https://doi.org/10.1016/j.cmet.2016.04.002). For one data set, we reanalysed the count matrix at GEO with accession number GSE85567 to get the effect sizes.

## Code availability

All code is available as open source R package scPower on github https://github.com/heiniglab/scPower and on Zenodo https://doi.org/10.5281/zenodo.5552753[99]. Code to reproduce the figures of the paper is provided in the package vignette. The repository includes a shiny app with a user-friendly graphical user interface, which is additionally available as a web server at http://scpower.helmholtz-muenchen.de/.

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

## Acknowledgements

We thank Thomas Walzthoeni for bioinformatics support provided at the Bioinformatics Core Facility, Institute of Computational Biology, Helmholtz Zentrum München. We thank Elisabeth Graf and Thomas Schwarzmayr for help in sequencing. We thank the BeCOME study team at the Max Planck Institute for Psychiatry, including the BioPrep core unit for their contribution to control individuals recruitment and characterizations, as well as collection of PBMCs. We thank Maren Büttner for insightful discussion and proofreading of the manuscript. H.L. is grateful for support by "ExNet-0041-Phase2-3 ("SyNergy-HMGU")" through the Initiative and Network Fund of the Helmholtz Association. CC is supported by a Banting Postdoctoral Fellowship. F.J.T. acknowledges support by the BMBF (grant # 01IS18036A and grant # 01IS18053A), by the Helmholtz Association (Incubator grant sparse2big, grant # ZT-I-0007) and by the Chan Zuckerberg Initiative DAF (advised fund of Silicon Valley Community Foundation, 182835). M.H. acknowledges support by the Chan Zuckerberg Foundation (CZF Grant #: CZF2019-002431). B.H. is supported by the Helmholtz Association under the joint research school "Munich School for Data Science—MUDS".

## Author contributions

K.T.S., B.H. and M.H. conceived the power analysis framework and analyzed the data. M.H., F.J.T., E.B.B. and H.L. designed the scRNA-seq experiment. E.B.B. planned the BeCOME study and recruited the study participants. C.C. and A.B. generated scRNA-seq data in PBMCs. K.T.S., B.H. and M.H. wrote the manuscript with input from all authors. All authors approved the final manuscript.

## Funding

## Competing interests

F.J.T. reports receiving consulting fees from Roche Diagnostics GmbH and Cellarity Inc., and ownership interest in Cellarity, Inc. and Dermagnostix. The other authors declare that they have no competing interests.
