## [Peer Review File · Nature Communications]

scPower accelerates and optimizes the design of multi-sample single cell transcriptomic studiesReviewers' Comments:

Reviewer #1:

Remarks to the Author:
see attached pdf

Review Schmid et al

In this paper the authors introduce an R-package for the design of single cell RNA-seq data for multi-sample analyses, such as they would occur in clinical settings involving multiple patients or for eQTL analyses. In such analyses effect sizes will often be small and experiments need to be designed to provide sufficient power. Here, the authors seek to solve this problem, and optimise the detection power for DE genes for a fixed cost by optimising the sequencing depth/ cell, cells/sample and the number samples.

To achieve this the authors require a set of priors including gene-wise fits of a negative binomial distribution (NBs) for the cell type of interest and a saturation curve of total UMIs vs total reads. There are also a bunch of other technical parameters that can be set, such as the doublet rate and the experimental costs and details of the chosen library preparation method and sequencing.

Next, the total counts per gene per individual are obtained using a cumulative NB based on the priors and the parameters of interest.

Furthermore, the user has to decide on reasonable expression detection cut-offs, i.e. in how many individuals does a gene have to be detected. Because it is an analytical solution, it is fast and it is probably a good first estimate for experimental design. However, the user has to be aware that estimates are not conservative. The model does not include errors and ignores real life problems like cell-type miss-assignment or that doublets might actually be mixtures of different cell types.

Furthermore, the eQTL DE analysis is not very well described, all important assumptions about the DE-settings and eQTLs are hidden in the Methods and difficult to understand. It appears that the main difference is that the group sizes for the DE-analysis is fixed and approximately equal, whereas the group sizes for which the DE analysis for eQTLs is based on a range of genotype frequencies assuming co-dominance, whereas the allele frequencies of the putative eQTLs are modelled as an (highly unrealistic) uniform distribution. This would be fine if the provided power analysis were conditional on the allele frequencies, but the marginals are not informative with such unrealistic settings.

Generally, I believe that also for the DE-design a flexible determination of group sizes would be helpful. I neither understood why only the eQTL detection power depends on the mean expression, shouldn't this also be true for DE-analysis?

Lastly, the default for a gene needing to be detected in 50% of the individuals is unreasonable in particular for eQTL analysis, assuming a gene is only detectable based on a 20% allele — this would be kicked out.

This leads to my main criticism: the lack of a batch effect in the model. If I understand correctly, the initial set of gene-wise NBs must contain all the variance expected, in the absence of the introduced DE-effects, all cells of the experiment fall nicely into this distribution. This is an unreasonable assumption. Especially with patient data batch effects are often unavoidable and need to be taken care of. Given that the authors have multiple individuals and batches, it should be possible for them to analyse whether one cell type is indeed homogeneous across batches or individuals and add the factor batch to the model.

The final straw for me not to recommend this study broke, when I played with the shiny implementation and found already the first result very puzzling and completely unexplainable: **When keeping the both reads/cell and the total number of cells constant, how can the detection power decrease with the sample size?** The only possibility I could think of is that the pseudobulk approach is deflating the variance for low sample sizes more, thus increasing the power. Did you ever try to keep the number of cells that are summed up for pseudobulk constant? In any case this result does not make any sense, and points to a serious problem at the core of the analysis.

All in all, this makes me mistrust the entire implementation and thus the paper, in particular since the focus is on multiple samples.

Reviewer #2:

Remarks to the Author:

The manuscript describes a toolkit for designing single cell RNA sequencing experiments and performing power analysis in detecting genes, that covers several interesting topics in designing single cell RNAseq experiments. A robust and precise tool serving these purposes is in great need in nowadays scRNAseq application fields. The idea of using pseudobulk in combination with existing power calculation tool is effective and excellent. However, there are some logic reasonings that may cause misunderstandings and a few technical comparisons that authors need to address. I have both major and minor comments as below:

Major comments:

1. Figure 1 is not very useful in helping readers to understand the algorithm. A better schematic illustrating the parameters and outputs of this tool is needed. This should list the exact parameters that are used in the algorithm.
2. This manuscript defined "A gene is called expressed with count > 10 or count >0 in more than 50% of the individuals", which is problematic. Different from bulk RNAseq experiments, a gene can be defined as "expressed" in an individual if it presents in certain % cells, for instance 15% of all cells within an individual. Requiring 50% individuals expressing this gene is not appropriate. The individual-level and group-level of gene detection should be stratified.
3. This is related to comment #2, setting a constant threshold, for instance count > 10 in pseudobulk is not preferred, because it will heavily depend on the number of cells. A total of read count =10 from a cell type with 1000 cells might be caused by technical noise; whereas count =10 in a small population with only 5 cells might indicate that almost all cells express this gene. Instead, a constant fraction of cells with count >0 might be a better alternate.
4. This relates to comment #2 again. A gene detected in any individual should be called as detected, which is to say that having more individuals is expected to provide better chance to detect a gene. The authors require 50% individuals to define a gene detection, which causes opposite results than this expectation. For instance, in the last piece of results for detecting rare cell types, with same cell type frequency and detection power, to detect 50 cells per cell type per individual, scPower calculates that 1390 cells per person are needed to be sequenced when sequencing 10 individuals; whereas more cells 1469 cells are required when sequencing 40 individuals.
5. Figure 4 and Figure 5 showed results from 10X and smartseq2 data respectively, which had discrepancy in several parameters. The authors need to analyze and discuss the difference between the two platforms and this difference should reflect in the algorithm parameters.
6. How about direct comparisons between more cells and more individuals with same budget? For instance, with 40K budget (let's say \$4K total cost per 10x library), should you sequence 10,000 cells per individual for 10 persons, or 5,000 cells per individual for 20 persons, or 2,500 cells per individual for 40 persons. It's better to make this type of serial comparison to provide direct impression to readers.

Minor comments:

1. The title of this article is misleading – the tool is designed for single cell RNAseq data, while single cell genomics refer to single cell DNA/genome studies that are different.
2. When the author claimed that their modeled results and observed results are very close in Figure 2, statistical tests are needed to make the statement.

3. The usage of very light colors reduced the readability, such as light pink, light grey, light yellow in Fig 4 and 5.

4. Figure 2 used the same set of colored lines for different annotations in same Figure that often cause wrong interpretation of the data.

Reviewer #3:

Remarks to the Author:

In this manuscript, Schmid et al. introduce scPower, a tool for power evaluation and sample size recommendation for single cell RNA-seq data, under the context of two-group differential expression (DE) tests for both DE genes and eQTLs. The authors adopted "pseudobulk" approach which is previously proposed by Dr. Mark Robinson, to calculate the detection power. However, the conclusion itself such as "the breadth is more preferable than the depth" is not unique.

Overall, the manuscript is well-written including the comprehensive literature reviews, and carries tests on different scenarios. However, I have some discretionary comments and critics for statistical modeling and assumptions.

Major:

1. The definition of detection power P_i is calculated as the product of expression probability (P_E) and detection power for DE (P_S). However, the author did not clarify why these two probabilities are independent. For example, simply image whether a gene expressed or not expressed follows a binomial distribution, then the statistical test will not become sensitive at the $P_E \sim 0.5$. Another conclusion from the bulk RNA-seq analysis also validated in scRNA-seq is that DE genes with higher expression values are easier to be detected than the genes with lower expression values. Given the fact that P_E is a cell-wise probability quantifying expression rate larger than a threshold (e.g., 10 or 0), which is also related to the gene expression level; therefore, P_E is not independent from P_S . The author should properly justify the assumption of independence for P_E and P_S using real data.

2. Related to comment (1), the definition of P_E is similar to the definition of "detection rate" which is dependent on the sequencer platforms as discussed in [1], how to optimize the minimum expression threshold (δ) to calculate the P_E is still not clear. Using $\delta=10,3,0$ universally might not be reasonable. A data-driven approach could be a better choice.

3. To my understanding, using this "pseudobulk" approach for power calculation can potentially bring batch effects for example each donor could possibly be defined as one batch (the sample variation can be confounded by batch effects). The author should address the concern of how to properly correct batches. Furthermore, Mark Robinson's 2020 bioRxiv paper indicates a similar issue.

4. The proposed method claims that using non-zero-inflation is sufficient to capture the pseudo bulk gene-wise distribution for different sequencers shown in (Table S4). However, it has been reported that the raw read counts from Smart-seq have zero-inflation, while Unique Molecular Identifiers (UMIs) counts can be adequately modeled by a negative binomial distribution (NB) [2-4]. The manuscript should address this choice in light of this debate and argue for the appropriateness of their choice based on various datasets.

5. It has been demonstrated that typical DE testing methodologies can suffer from spurious results that derive from clustering the data first and then running DE testing methods on the resulting clusters. The authors also fall into this camp that applying clustering first and then performing DE tests (lines 150-151). This will further hurt the multiple test adjustment.

6. Followed by comment (5), the authors use two approaches FDR and FWER for DE/eQTL genes. What does the distribution of the original p values obtained from the DE tests look like? For FWER approach, why use threshold $0.05/(E*10)$? How many significant eQTL genes remain if using $E*100$, $E*1000$? Also, for FDR approach, is there any advantages to adopt the Jung's formular and how it is related to your equation? From the description (line 877-880), the d' is an interval, then r_{-1} calculate the sum of marginal power based on different d' ? Why r_{-1} calculate the expected number of significant DE DEGs/eQTLs?

7. Since the effect size (lfc) or the cell-type-specific prior is computed from the pilot dataset, the authors should evaluate the similarity between the simulated data and the reference data in order to adopt the effect size for calling DE and ranking genes. Because the # of cells per cell type could impact the effect sizes. Specifically, the called DE gene ranks should be preserved like the real DE gene ranks. Moreover, since scPower uses effect size to determine DE genes, it is not fair to compare with powsimR and muscat with respect to running time for they use different ways to call DE genes in the Figure3.

8. There is only a small section talking about rare cell type detection, and it seems independent from the overall manuscript because it is not related to the theme of detection power.

9. There is an assumption made in the model that cell types present, and their proportion in the sample will be replicated in future experiments (otherwise the power calculation is not appropriate). This is a very strong assumption, especially considering that cell types are inferred from clustering, which can be affected by variability in the composition, depends on the resolution used to determine clusters etc. The manuscript should include (1) an analysis of the robustness of the method as proposed to variability in cell composition; (2) if necessary, an extension to the proposed method to address this.

10. Since the authors mentioned the feasibility of extending the two-group test into multi-group test via this pseudobulk approach, it is better to showcase the investigation result on one of the PBMC datasets.

Minor:

11. In line 1016, n_s is the quotient of two quantities.

12. Some R package names are in italic font some are not.

13. scPower calculates the gene-wise means and use splatter mixture model to capture the cell-wise distribution of the means corresponding to all the genes. Why not directly use splatter mixture model to capture the pseudobulk counts per cell type? Is there a merit of applying NB first?

Reference:

[1] Hicks, Stephanie C., et al. "Missing data and technical variability in single-cell RNA-sequencing experiments." *Biostatistics* 19.4 (2018): 562-578.

[2] Grün, Dominic, Lennart Kester, and Alexander Van Oudenaarden. "Validation of noise models for single-cell transcriptomics." *Nature methods* 11.6 (2014): 637-640.

[3] Svensson, Valentine. "Droplet scRNA-seq is not zero-inflated." *Nature Biotechnology* 38.2 (2020): 147-150.

[4] Townes, F. William, et al. "Feature selection and dimension reduction for single-cell RNA-Seq based on a multinomial model." *Genome biology* 20.1 (2019): 1-16.

Point by point response

First of all, we would like to thank all three reviewers for their valuable comments and for having taken the time to review our work. We found the reviewers' comments very helpful and believe that the changes we have made based on their feedback have strengthened our study. Overall, we took the comments from the reviewers as indicators showing us what parts of our manuscript we had to rephrase more clearly and where to add further explanations. We also improved the visualizations to make interpretation of the plots more intuitive and avoid misinterpretations of the results. Along the same lines we made navigation of the website more user-friendly. Additionally, we completely redid Figure 1 and are convinced that it now provides a better overview of the workflow.

We have introduced additional user flexibility in response to several comments about the type and choice of thresholds. We addressed the concern that batch-effects or differential cell type abundance might impact the power calculations and show that power estimates remain accurate. Furthermore, we show how scPower can be extended for more complex designs and provide instructions to the users in a detailed package vignette. In the following, we provide point by point responses for all comments.

Reviewer comments are marked in green and italic font.

Responses are typeset in regular font.

Edits in the main text are marked in blue.

Reviewer #1 (Expertise: statistical methods for the analysis of scRNAseq):

In this paper the authors introduce an R-package for the design of single cell RNA-seq data for multi-sample analyses, such as they would occur in clinical settings involving multiple patients or for eQTL analyses. In such analyses effect sizes will often be small and experiments need to be designed to provide sufficient power. Here, the authors seek to solve this problem, and optimize the detection power for DE genes for a fixed cost by optimizing the sequencing depth/ cell, cells/sample and the number samples.

To achieve this the authors require a set of priors including gene-wise fits of a negative binomial distribution (NBs) for the cell type of interest and a saturation curve of total UMIs vs total reads. There are also a bunch of other technical parameters that can be set, such as the doublet rate and the experimental costs and details of the chosen library preparation method and sequencing.

[1] Next, the total counts per gene per individual are obtained using a cumulative NB based on the priors and the parameters of interest. Furthermore, the user has to decide on reasonable expression detection cut-offs, i.e. in how many individuals does a gene have to be detected. Because it is an analytical solution, it is fast and it is probably a good first estimate for experimental design. However, the user has to be aware that estimates are not conservative. The model does not include errors and ignores real life problems like cell-type miss-assignment or that doublets might actually be mixtures of different cell types.

We thank the reviewer for highlighting the advantages of our model in particular its ability to compare and choose optimal experimental design options. We disagree with the reviewer's

opinion that our method is not conservative and does not include errors. We do model the uncertainty of the measurements in multiple aspects, such as the technical and biological variation of the single cell gene expression measurements. These are captured in the negative binomial model on the single cell level. The interindividual variability is captured by the negative binomial model on the pseudo bulk level.

We also address real life problems for instance by estimating the doublet fraction based on the overloading of the lane. This fraction of cells is then excluded in the analysis, adjusting the power estimation. We agree with the reviewer that the doublets will be both mixtures of cells of different donors and mixtures of cells of different cell types for the same donor. For that reason, we estimate the doublet fraction in our pilot data set by combining two complementary methods: 1) demuxlet, which identifies doublets originating from donor mixtures based on genetic variants, and 2) scrublet, which identifies doublets originating from cell type mixtures for the same donor. For cost optimization of 10X experiments, the doublet rates are modeled as a function of the number of cells per lane, based on reference values provided by 10X. If the user wishes to take a more conservative approach, they are free to set their own estimates for the doublet rate in our web tool or the R package, where it is passed as a parameter to our model.

We agree with the reviewer that cell type misassignment could in theory also affect the analysis. Recent publications already investigated the effect of experimental parameters on the quality of cell type assignments (Mandric et al. 2020; Heimberg et al. 2016). Both showed that in practice an accurate cell type assignment is already possible at a very low read depth, e.g. Mandric et al estimate a misclassification rate of at most 4%. Due to this low reported error rate we deemed this issue negligible and decided not to consider potential cell type misassignment in our model.

[2] Furthermore, the eQTL DE analysis is not very well described, all important assumptions about the DE-settings and eQTLs are hidden in the Methods and difficult to understand. It appears that the main difference is that the group sizes for the DE-analysis is fixed and approximately equal, whereas the group sizes for which the DE analysis for eQTLs is based on a range of genotype frequencies assuming co-dominance, whereas the allele frequencies of the putative eQTLs are modelled as an (highly unrealistic) uniform distribution. This would be fine if the provided power analysis were conditional on the allele frequencies, but the marginals are not informative with such unrealistic settings.

There are multiple differences between the DE and eQTL analysis. Following the reviewer's suggestion, we have restructured the results section "scPower models the power to detect differentially expressed genes and expression quantitative trait genes" (page 10, line 325-329) and amended the manuscript (see page 11, line 336-340) to highlight the important differences between DE and eQTLs more clearly. As the reviewer pointed out correctly, one is the way the groups are defined: for the DE analysis the groups are based on a selected category such as healthy vs disease, while the eQTL groups are defined based on the genotypes (i.e. additive wrt minor allele count). The most important difference however, is the statistical testing procedure. For the DE analysis, we calculate the power based on negative binomial regression and for the eQTL analysis, based on linear models on the transformed read counts, as they are typically used due to their efficiency in performing very large numbers of statistical tests. This leads also to different effect size specifications, fold changes in the DE case and R-squared values in the eQTL case.

In general, the eQTL power analysis is not based on the allele frequency directly, but on the R-squared values, which combine the allele frequency (and thus the variance of the predictor variable) and the beta value (for the predictor) in the linear model (added now also to the manuscript in page 11, line 337-340). The R-squared values are taken as priors from published eQTL studies or they are simulated to have a similar distribution as the R-squared values from published studies (see also section “Reviewer 3, comment 6”). In the general case (expression levels high enough) the power calculation is done analytically based on F-tests dependent on the effect size (R-squared value), the sample size and the significance threshold, but independent of the mean gene expression level. This power calculation assumes that the residuals are i.i.d. normally distributed. For large enough count values, it has been shown that normalized log transformed counts have a constant variance independent of the mean value and can be analyzed with linear models (Law et al. 2014).

Regarding the concern of the reviewer that the allele frequencies are modelled by an uniform distribution, we want to clarify our approach. We use simulations to determine the eQTL power for genes with small mean values, i.e. only very few non-zero counts, as the normalization might not be effective (Supplementary Figure S12). Only in these cases, the allele frequency is specifically modelled.

During this simulation, we need to break down the R-squared values back to the allele frequency and the beta value. The reviewer correctly pointed out that we use uniform distribution for the eQTL allele frequency. We approximately observe this distribution for eQTL SNPs in the blueprint data (Figure 1) and thus consider it appropriate. While an uniform distribution is not observed when looking at allele frequencies of all genetic variants across the genome (which is skewed towards more rare variants), the shape of the distribution might be explained by the fact that we are only looking at variants which are eQTLs (usually only assessed for variants above a certain minimal minor allele frequency).

Figure 1. Minor allele frequency in the Blueprint eQTL studies for significant eQTLs.

[3] Generally, I believe that also for the DE-design a flexible determination of group sizes would be helpful.

We would like to thank the reviewer for this valuable suggestion. We have extended our DE model to now calculate the power also for imbalanced groups by adding a parameter that describes the ratio between both groups (number cases / number controls). This parameter is also available on our updated website, if the DE scenario is selected. As expected, imbalanced groups reduce the power (Figure 2). This effect is stronger for lower power settings and with more extreme imbalance.

Figure 2: Evaluation of the effect of imbalanced group sizes. The power for parameter combinations of Main Figure 3A (each point representing one combination) was recalculated assuming instead of balanced groups (x-axis and default value in the paper) a sample size ratio between both groups of 1/3 and 1/7 (y-axis, color-coded).

[4] I neither understood why only the eQTL detection power depends on the mean expression, shouldn't this also be true for DE-analysis?

This is correct and has been clearly stated in the manuscript: both the DE and the eQTL detection power depend on the mean. For the DE power analysis this mean dependence is naturally resulting from using the negative binomial power analysis framework. We stated this for example here in the manuscript:

“In addition, both the expression probability and the DE/eQTL power depend on the mean μ and dispersion ϕ of expression levels of gene i .” (Results, page 7, line 207-208)

“Building on our expression probability model, we can assess the DE/eQTL power of the expressed genes using existing analytical power analysis tools that have been established for bulk sequencing data. They estimate the power to detect an effect of a given effect size depending on the sample size, the gene mean expression level and the chosen significance threshold.” (Results, page 10, line 321-325)

For the eQTL case the power analysis is based on linear models, where the mean dependence is not naturally given (see also response to comment 2). This is why we had provided more explanation of eQTL mean dependency in the text (page 13 line 397-403). We hope that this

becomes more clear now that we restructured the section “scPower models the power to detect differentially expressed genes and expression quantitative trait genes” (page 10-13) in response to comment 2 of the reviewer.

[5] Lastly, the default for a gene needing to be detected in 50% of the individuals is unreasonable in particular for eQTL analysis, assuming a gene is only detectable based on a 20% allele — this would be kicked out.

We thank the reviewer for this comment, we agree that a threshold of 50% is probably too strict as default in the eQTL case. We therefore updated the new Main Figure 4 (before in the supplement) and Main Figure 5 (former Main Figure 4) using a population threshold of 9.5% (based on minor allele frequency as explained below) instead of 50% in the eQTL scenarios. In general, we implemented all thresholds in a flexible way so that the users can decide themselves how to choose the thresholds. The thresholds depicted in the Figures are always examples. Following the approach of edgeR (Y. Chen, Lun, and Smyth 2016), we suggest setting the expression percentage based on the smaller group size, which results in a percentage threshold of 50% for a balanced DE design. For the eQTL analyses, we would consider the expression based on the minor allele frequency (MAF), so that the gene is at least expressed in the heterozygotes, leading to a threshold of $2 \cdot \text{MAF} \cdot (1 - \text{MAF})$. When analyzing an eQTL with a minor allele frequency of 0.05, which is a common lower threshold for genetic variants tested for associations, this corresponds to a percentage threshold of 9.5%.

Comparing the results of Main Figure 5 with a threshold of 50% to the results based on the new cutoffs, we see that the lower threshold leads to a higher detection power (Figure 3). In many cases, a slightly lower number of cells is selected, probably as the expression probability is in general higher with a lower expression cutoff, whereas the optimal sample size is increased. The read depth remains low throughout and hence the general recommendation of shallow sequencing of more cells remains valid.

Figure 3. Effect of population level cutoff on Main Figure 5. Comparing the old version of main Figure 5 with a population cutoff of 0.5 and the new version of 0.095 showed that the detection power increases slightly with the lower cutoff, but the general trends for the parameters remain the same.

The other reviewers had additional questions regarding the thresholds and we would like to refer to reviewer #2 comment 2 and 4 and reviewer #3 comment 2 for an in-depth discussion of the thresholds. To summarize them, we now extended our method to take alternatively absolute thresholds (a gene needs to be expressed in a specific number of individuals) and to select the expression threshold that maximizes the overall detection power. Both extensions further expand the user flexibility when choosing the thresholds.

[6] This leads to my main criticism: the lack of a batch effect in the model. If I understand correctly, the initial set of gene-wise NBs must contain all the variance expected, in the absence of the introduced DE-effects, all cells of the experiment fall

nicely into this distribution. This is an unreasonable assumption. Especially with patient data batch effects are often unavoidable and need to be taken care of. Given that the authors have multiple individuals and batches, it should be possible for them to analyse whether one cell type is indeed homogeneous across batches or individuals and add the factor batch to the model.

We would like to thank the reviewer for raising this point and we agree that we need to consider batch effects in our model. We have added the results of the following considerations to the manuscript. Batch effects can affect three different steps in our workflow; differences might occur for batches within the pilot data set for the expression prior, between the pilot data set and the planned experiment that we are estimating the power for and within the planned experiment itself.

In the first scenario, during the generation of our expression priors, the selection of the pilot data is pivotal. We now specified in the manuscript that the pilot data represents controls without strong DE effects, that cover the natural inter-sample variability (equivalent to the requirements of simulation-based methods such as powsimR) (page 8, line 244-245). Although batch effects in our pilot data set were not visible on the UMAPs (Figure 4) we took precautions, as the UMAPs are not directly quantifying batch heterogeneity. To prevent batch effects, we had fitted the single cell negative binomial distributions separately for each run. We apologize for not having specified this in more detail in the methods and we have now amended the manuscript to describe this more clearly (page 23, line 793-795). Next we fitted the gamma mixture models also separately for each run. Finally we combined the runs in the step where we estimate the dependence of the parameters of the gamma mixture on the number of mapped reads. Nevertheless, the negative binomial model is also able to account for heterogeneity between cells of the same cell type across runs, which would result in higher dispersion. To quantify the congruence of the distribution fits between batches, we investigated the mean and dispersion results. First we compared how well the negative binomial means matched across the batches. We saw that it was high with correlation between means > 95% for all runs and cell types (exemplarily shown for two runs in Figure 5A). Second, we estimated the function describing the mean to dispersion relation separately for both runs and also on the combined data set. As expected, we indeed observed a slightly higher level of dispersion in some cell types when fitting these functions on the combined data (Figure 5B). Higher dispersion estimates lead to a conservative underestimation of power. While this is not optimal this conservative behaviour is preferred. In conclusion, the optional scenario is to generate pilot data in a single batch. If batches are present, the expression model should be fit separately and the parameters should be combined subsequently. Lastly, if unknown batch effects are present in the pilot data, this will result in conservative power estimates.

Figure 4. UMAP of training data set PBMC1 colored by cell types (A, as in Figure S2A) and by batches (B).

Figure 5. Comparing negative binomial fits between runs (batches) separately for each cell type, considering both the mean (A) and the dispersion functions (B). The mean values (logarithmized means) of Run 1 and Run 2 were plotted against each other (A). For the dispersion functions, additionally to the functions fitted for Run 1 and Run 2, a function fitted on the combined matrix of both runs is shown (B).

Regarding the second scenario, differences between the pilot data set and the planned experiment are a challenge not unique to analytic methods but equally concern simulation-based tools such as powsimR and muscat. In general, disparities should be prevented by choosing the pilot data set for the prior as close to the planned experiment as possible and by ensuring that the pilot data set was measured with the same single cell technology. We show that using expression priors from a training set can successfully predict expression in a validation dataset (Main Figure 2C) even though they were generated by different labs and analyzed with different workflows. For this reason, we are confident that a transfer of expression priors measured in different environments will also be successful in other cases.

The third scenario, differences within batches in the analysis data set, can be overcome by non-confounded experimental design and by using covariates in the model (see also reviewer #3, comment 3). Equivalent to classical bulk DE and eQTL analysis, covariates can be added to the model of single cell RNA-seq analysis to account for different batches (W. Chen et al. 2020). This is even possible for unknown confounding variables based on latent factor analysis (Stegle et al. 2012; Buettner et al. 2017) and increases the power compared to non-batch corrected analysis (Hernández, Steyerberg, and Habbema 2004; Stegle et al. 2012; Kahan et al. 2014).

To measure any reduction of power due to batch effects, we analyzed how well edgeR performs under batch effects of different strengths using powsimR and how much our scPower power values deviated from the power values observed in this simulation including batch effects. We simulated the batch effect in powsimR by introducing different batch log fold changes to randomly sampled 20% of the genes, assigning the batches under the assumption of non-confounded experimental design. This means that we generated 2 batches with 50% cases and 50% controls each. We observe that especially when the batch effects are higher than the effect sizes of the DE genes (mean effect size: absolute log₂ fold change of 2.8) the power is dropping drastically without batch correction in the covariates. As a side note, the differences between scPower and powsimR in the condition without batch effects match the results shown in the main manuscript (Main Figure 3C-E and page 13-14, line 433-445) and can be explained by different modelling assumptions (in these cases muscat usually estimates slightly higher power than scPower). Importantly, taking the batch as a covariate leads to the same power as in the simulation cases without any batch effects, even in cases with high batch effect sizes (Figure 6).

We added a short section on this analysis to our manuscript (page 14, line 446-455). These results demonstrate that our current analytic method is fully capable to deal with batch effects in properly designed experiments. We now emphasize in the manuscript that it is critically important to have a proper design of the experiment with non-confounded batches. This allows for a proper batch correction during the analysis and the power estimations with our method remain accurate.

Concluding, batch effects can be dealt with if they are properly treated within the pilot and analysis data set and to some degree avoided altogether if the pilot data set is chosen as close to the analysis data set as possible and generated as a single batch.

Figure 6. PowsimR simulation results for number of expressed genes (A), DE power (B) and overall detection power (C) for data with batch effects of different log fold change (x-axis). The effect was evaluated for different parameter combinations of sample size - number of cells per sample (see facet header). The red lines visualize the prediction of scPower for this parameter combination. The power drops considerably if the log fold change of the batch effects is not corrected using covariates in the edgeR (violett), while with correction (green) it stays at the same level as without batch effects (blue).

[7] The final straw for me not to recommend this study broke, when I played with the shiny implementation and found already the first result very puzzling and completely unexplainable: When keeping the both reads/cell and the total number of cells constant, how can the detection power decrease with the sample size? The only possibility I could think of is that the pseudobulk approach is deflating the variance for low sample sizes more, thus increasing the power. Did you ever try to keep the number of cells that are summed up for pseudobulk constant? In any case this result does not make any sense, and points to a serious problem at the core of the analysis. All in all, this makes me mistrust the entire implementation and thus the paper, in particular since the focus is on multiple samples

Study parameters

General parameters

Study type DE study eQTL study

Cell type

Cell type frequency

Reference study

Total budget

Parameter grid

Samples (min)

Samples (max)

Reads (min)

Reads (max)

Steps

The figure represents the detection power that can be gained with each parameter combination of cells per individual and read depth. The third parameter, the sample size, is defined uniquely by the other two parameters and the overall experimental budget.

To update the plots with the currently set parameter combinations on the left, please click the Calculate button. The calculation for a specific parameter combination can take up to 1-2 minutes.

Figure from reviewer

We apologize for this misunderstanding. Most likely the reviewer did not interpret our shiny figure as intended, since it is not possible to simultaneously keep both the read depth and number of cells constant within the shiny app. The plot quoted here from the reviewers comments shows the results of the cost optimization. In this analysis the budget is fixed, two of the three free experimental design parameters are set to values on a grid and the third one is found by optimizing the power. In the setting that the reviewer tried out the budget is fixed, the sample size and the read depth are set to specific values in the grid and the number of cells is optimized to achieve highest power. Therefore, at the same read depth and with increasing number of samples the number of cells has to decrease to stay in the budget, which reduces the power.

To enhance the interpretability of the results of our tool, we now visualize the third cost determining factor through the circle size (Figure 7), so that it is no longer hidden in the mouseover text. In general, we improved the documentation of the website, enhanced the design and functionality to make it more user-friendly.

Figure 7. New layout of our shiny server

We visualize the difference between an analysis where cell numbers are determined to optimize power for a fixed budget (as done in the app) and an analysis with constant cell numbers (as proposed by the reviewer) in Figure 8. The line plots show the overall detection power for a read depth of 100,000 and increasing number of samples from 10 to 50, so representing the bottom row of the screen shot from the reviewer. In the shiny app, the red line (“budget restricted cell numbers”) in Figure 8A is visualized. To stay within the overall budget of 50,000€, the number of cells must be drastically decreased for a higher sample size. This leads then also to a lower overall detection power (Figure 8D, red line), matching with our overall observations that the number of cells is a major determinant for the power. This reduced overall detection power is mainly driven by the lower expression probability for fewer cells (Figure 8B, red line; compare also to Main Figure 3), while the DE power stays nearly constant (Figure 8C, red line). If we kept the number of cells constant (Figure 8A, blue line), a higher sample size would of course lead to a higher overall detection power (Figure 8D, blue line) due to higher DE power (Figure 8C, blue line). We extended our explanation in the manuscript on how each parameter influences the expression probability, the DE/eQTL power and the overall detection power (Main Figure 4 and page 15, line 498-511). In conclusion, the suspected issue with the code was in fact a misreading of the visualization for which we apologize; the number of cells is decreased due to a fixed budget which reduces the overall power.

Figure 8. Number of cells (A), expression probability (B), DE power (C) and overall detection power (D) for a read depth of 100,000 and a sample size between 10-50 (x-axis). The version “budget restricted cell number” (red line) is the variant visualized in our shiny app, where the experimental cost needs to be below a certain budget. In contrast, the version “constant cell number” shows a budget independent variant with constant number of cells independent of the sample size.

Reviewer #2 (Expertise: biostatistics, scRNASeq analysis):

The manuscript describes a toolkit for designing single cell RNA sequencing experiments and performing power analysis in detecting genes, that covers several interesting topics in designing single cell RNAseq experiments. A robust and precise tool serving these purposes is in great need in nowadays scRNAseq application fields.

The idea of using pseudobulk in combination with existing power calculation tool is effective and excellent. However, there are some logic reasonings that may cause misunderstandings and a few technical comparisons that authors need to address. I have both major and minor comments as below:

Major comments:

1. Figure 1 is not very useful in helping readers to understand the algorithm. A better schematic illustrating the parameters and outputs of this tool is needed. This should list the exact parameters that are used in the algorithm.

We agree that Figure 1 was not ideal to visualize the algorithm. We completely restructured the figure and added the user defined parameters that are used during the modelling. Associated with this, we also updated Figure 2A so that the color coding and used terms match with Figure 1. We hope that the reviewer agrees that the figure is now more intuitive and presents in more detail which parameters are used in each step of the algorithm.

2. This manuscript defined “A gene is called expressed with count > 10 or count >0 in more than 50% of the individuals”, which is problematic. Different from bulk RNAseq experiments, a gene can be defined as “expressed” in an individual if it presents in certain % cells, for instance 15% of all cells within an individual. Requiring 50% individuals expressing this gene is not appropriate. The individual-level and group-level of gene detection should be stratified.

We want to thank the reviewer for his comments related to our threshold which led to a more extensive evaluation of the different threshold options. First, we want to point out that the chosen thresholds in our manuscript are specific examples and all expression thresholds in our models can be flexibly defined by the user. As discussed in the response to Reviewer #3 comment 2, setting a threshold balances the detection power and the false positive rate. Following the suggestion in that comment, we implemented an additional option to select the expression threshold that maximizes the overall detection power in a data driven way, which however leads to potentially higher false positive rates (Soneson and Robinson 2018). For this reason, we give the users the possibility to set the thresholds themselves according to their experiment and preferences.

The first part of the comment - setting a percentage cutoff (expressed in x% of the cells of an individual) instead of an absolute cutoff (expressed with >x counts in the pseudobulk) on the individual level - will be covered in detail in the response to comment 3.

Regarding the second part of the comment: to define a gene as expressed in a population if it is expressed in a certain fraction of individuals, is a common strategy in bulk. For example edgeR suggests exactly this kind of filtering (Y. Chen, Lun, and Smyth 2016) and also implements a function in their R package called “filterByExpr()” (Y. Chen et al. 2021). As the edgeR DE analysis is outperforming other methods for single cell inter-individual DE analysis (Crowell et al. 2020), we expect that also the same filtering strategy is applicable. Following this, we suggest setting the expression percentage based on the smaller group size, which results in a percentage threshold of 50% for a balanced DE design. For the eQTL analyses, we would consider the expression based on the minor allele frequency (MAF), so that the gene is at least expressed in the heterozygotes, leading to a threshold of $2 \cdot \text{MAF} \cdot (1 - \text{MAF})$. When analyzing an eQTL with a minor allele frequency of 0.05 this corresponds to a percentage threshold of 9.5%. We updated Main Figure 4 and Main Figure 5 based on this threshold for

eQTL experiments and explicitly explained our approach in the manuscript (page 15, line 491-497) and added a new Methods section (page 25-26, line 869-891).

Furthermore, when performing inter-individual comparison of gene expression, a gene that is expressed in only one individual has probably also a low statistical power and will only increase the multiple testing burden without contributing relevant results.

Responding to comment 4, we extended our filtering options to include absolute thresholds on the population level (a gene needs to be expressed in a fixed number of individuals). See response to comment 4 for a detailed discussion of the different options.

A large-scale analysis of different filtering strategies is beyond the scope of our manuscript (different previous studies came to contradicting results (Soneson and Robinson 2018; Vieth et al. 2019)) and evaluating filtering strategies is also not the goal of our tool. Instead, we want to give the users the freedom to choose their own cutoffs. If the users prefer not to do population-level filtering, they can set the population-level threshold to 0. Then the gene needs to be expressed in at least one individual to be called as expressed, as suggested by the reviewer.

3. This is related to comment #2, setting a constant threshold, for instance count > 10 in pseudobulk is not preferred, because it will heavily depend on the number of cells. A total of read count =10 from a cell type with 1000 cells might be caused by technical noise; whereas count =10 in a small population with only 5 cells might indicate that almost all cells express this gene. Instead, a constant fraction of cells with count >0 might be a better alternate.

We thank the reviewer for the input and want to explain the advantage of an absolute cutoff on pseudobulk over using fractions of cells before we show additional results based on the fraction of cells. Using a fixed pseudocount threshold leads to an increase of expressed genes with higher numbers of cells. This is in line first with the empirical observation in many sequencing applications (e.g. also ATAC), that more starting material yields more precise measurements. Second, also the statistical intuition (variance of the mean decreases with increasing number of cells) would imply that more cells are beneficial for the accuracy of the expression measurements and thus one would expect to be able to call more genes “expressed” with more cells. The key drawback when applying the cutoff on the fraction of cells criterion is that there is no chance to increase the number of detected genes per cell type by increasing the number of measured cells (see empirical results below). This behavior is conflicting the empirical and statistical intuition explained above and contradicts the recommendation of shallow sequencing of more cells, which is also given by other authors (Mandric et al. 2020; Heimberg et al. 2016).

Nevertheless, following the reviewers suggestion and based on the single cell expression distributions already in place, we implemented a second strategy to define the expression cutoff on the individual level. Additionally to the method of an absolute cutoff in the pseudobulk (count > x), the users can now alternatively choose that the gene needs to be expressed in x% of the cells for an individual and cell type with count > 0. We validated that modeling based on the single cell expression distributions matches the observed expressed genes according to the new definition (Figure 9), equivalent to Main Figure 2B.

Figure 9. Expression model when applying a percentage cutoff (expressed in x% of the cells) instead of an absolute cutoff ($> x$ counts in pseudobulk) for the individual level threshold. Calculated for the same data set as in main Figure 2B-C. The observed number of expressed genes (solid lines) closely match the ones estimated with scPower (dashed lines). Curves were calculated once with a population cutoff of expressed in $> 50\%$ of the individuals (blue lines) or once without any population level filtering (red lines).

When applying a percentage cutoff on the number of cells with count > 0 , a similar effect occurs as observed by the reviewer in comment 4: a higher number of cells leads to a lower number of expressed genes according to this definition. This is especially visible when applying no population level filtering (population cutoff $> 0\%$), as suggested in comment 2 of the reviewer. Consequently, also the optimization results would look completely different, as increasing the number of measured cells does not help to increase the number of expressed genes (see Figure 10). So in this scenario, the optimal parameter combination (without considering the budget) would be measuring 100 cells in 20 samples, while with the absolute cutoff it was 30,000 cells in 20 samples.

In conclusion, while we now offer users the possibility to apply filtering based on a percentage threshold for their studies, we argue that absolute count thresholds are preferred and we will keep our initial filtering approach in the manuscript.

Figure 10. Variant of Main Figure 3A with a percentage cutoff: a gene is defined as expressed if it is expressed in at least 10% of all cells in at least 1 individual.

4. This relates to comment #2 again. A gene detected in any individual should be called as detected, which is to say that having more individuals is expected to provide better chance to detect a gene. The authors require 50% individuals to define a gene detection, which causes opposite results than this expectation. For instance, in the last piece of results for detecting rare cell types, with same cell type frequency and detection power, to detect 50 cells per cell type per individual, scPower calculates that 1390 cells per person are needed to be sequenced when sequencing 10 individuals; whereas more cells 1469 cells are required when sequencing 40 individuals.

This refers again back to the question of how to best set the expression thresholds. As explained in response to comment 2, we would suggest using a group level threshold in line with the edgeR suggestion, but the tool is flexible allowing for user defined choices. The group level filtering has the advantage that it reduces the multiple testing burden by focusing on genes that are more likely to harbor detectable significant effects (see also comment 2). Regarding the concerns of the reviewer, we agree that requiring expression in a higher fraction of individuals can lead to a lower expression probability and therefore also lower overall detection probability for a higher sample size (Figure 11). The connection between the population level expression probability and the sample size is however more complex. The population level expression probability depends on the individual level expression probability (shown in the different panels) and the fraction of individuals threshold (colors in Figure 11). For genes with a low individual level expression probability and a high fraction of individuals threshold, a higher number of samples leads to a decrease of the population level expression probability (e.g. for an individual expression probability < 0.5 and a fraction of individuals threshold > 0.5). In contrast with individual level expression probability of 0.5 or 0.7 and a fraction of individuals threshold of > 0.5 , a higher sample size leads to higher population-level expression probability.

Figure 11. Expression probability (y axis) dependent on sample size (x axis), chosen expression cutoff (expressed in x% of the individuals, color) and expression probability of the gene in one individual (panel title).

To make our model more flexible, we extended the population level filtering to additionally include absolute thresholds. This means that users can now specify either that the gene needs to be expressed in a certain percentage of individuals or in a certain absolute number of individuals. The absolute threshold avoids the drawback of the percentage threshold that a higher sample size might lead to a reduction of the expression probability (see Figure 12). Of course, the user is free to set a threshold of 0 and skip the filtering on the population level completely (in this case the gene needs to be expressed in at least one individual) or use the data driven approach to optimize power (see Reviewer #3 comment 2).

Figure 12. Variant of Main Figure 3A with an absolute expression threshold on the population level: a gene is defined as expressed if it is expressed with more than 3 counts in more than 2 individuals.

5. Figure 4 and Figure 5 showed results from 10X and smartseq2 data respectively, which had discrepancy in several parameters. The authors need to analyze and discuss the difference between the two platforms and this difference should reflect in the algorithm parameters.

Several small adaptations were implemented in our model to consider the platform-specific characteristics for Smart-seq2. We have rewritten the specific section in the manuscript to explain these better (page 19, line 583-588). In general, the main changes of the Smart-seq2 model compared to the 10X Genomics model are 1) to include the transcript length in the size normalization factor of our count model combined with a gene length normalized expression threshold to correct the gene-length bias, 2) to model the doublet rate as a constant factor (in contrast to variable rates in 10X dependent on overloading of the lanes) and 3) a platform-specific cost function with default costs typically for a Smart-seq2 experiment.

This leads to different optimal parameters and differences in reachable power for Smart-seq2 compared to 10X Genomics in Figure 5 and Supplementary Figure S18 (formerly Figure 4 and Figure 5). The common x-axis between both figures covers experimental budgets between 5,000€ and 100,000€, for which each the optimal experimental parameters are identified.

The biggest difference between the platforms that also creates the difference in the plots is that Smart-seq2 can not cost-efficiently process the same large number of cells as droplet based assays can. As we documented in Supplementary Table 5, we estimate the library preparation costs per cell for 10X Genomics between 0.05-0.12€ (dependent on how many cells were loaded per lane) and for Smart-seq2 at 13.00€. The sequencing costs stay the same.

Showing the effect for an example: if we take the prototypic DE scenario with low effect sizes and uniform rank distribution combined with a budget of 50,000€, the optimal parameter combination for 10X Genomics would be measuring a sample size 46 with 9000 cells per individual and a read depth of 20,000. The same parameter combination would cost

5,410,388€ for Smart-seq2. Instead, the optimal parameter combination within the budget contains measuring 36 samples with 100 cells per individual and a read depth of 200,000 reads per cell. However, when taking the optimal parameter combination of 10X Genomics from the example and calculating the power of both technologies, the Smart-seq2 experiment would even get a higher detection power with 60.9% compared to 41.9% for the 10X Genomics experiment (taking the same prototypic priors, but different cell types, as we have PBMCs for 10X Genomics and pancreas for Smart-seq2).

As we now explicitly state in the manuscript (page 19, line 600-603), Smart-seq2 experiments are not less powerful per se, but the significantly higher cost in the multi-sample setting leads to less powerful designs when restricting the budget. This allows only to measure much fewer cells, although a high number of samples and cells is often beneficial. We would argue that Smart-seq2 is a very valuable technology for analyses requiring high quality full-length transcripts, such as differential alternative splicing and when lowly expressed genes are of interest. However in the settings we evaluated it is often too expensive for larger multi-sample studies.

6. How about direct comparisons between more cells and more individuals with same budget? For instance, with 40K budget (let's say \$4K total cost per 10x library), should you sequence 10,000 cells per individual for 10 persons, or 5,000 cells per individual for 20 persons, or 2,500 cells per individual for 40 persons. It's better to make this type of serial comparison to provide direct impression to readers.

Indeed, finding the optimal design for a given budget is the main application for scPower and was not sufficiently highlighted in our first submission. We rearranged part of our manuscript to move Supplementary Figures S15 and S16 to the main text (and put former Figure 5 to the supplement instead) and added text (page 15, line 487-511) describing the different parameter options for a given budget better.

Minor comments:

1. The title of this article is misleading – the tool is designed for single cell RNAseq data, while single cell genomics refer to single cell DNA/genome studies that are different.

We agree with the reviewer that our title could be improved and changed it to: scPower - Fast design optimization to maximize power in multi-sample single cell transcriptomics studies

2. When the author claimed that their modeled results and observed results are very close in Figure 2, statistical tests are needed to make the statement.

We thank the reviewer for this suggestion and added **Supplementary Table S4** (see also below) plus references in the text and Figure 2 itself to statistically support the high quality of our fits in Figure 2. We assessed the difference based on Pearson correlation, reporting the r^2 and the p-value of the correlation. All estimations showed highly significant $r^2 > 0.9$.

Run	Technology	UMI cutoff > 0		UMI cutoff > 10	
		r ² value	p-value r	r ² value	p-value r
PBMC1 - Run 1	10X Genomics	0.982	9.11e-21	0.997	1.12e-29
PBMC1 - Run 2	10X Genomics	0.990	2.01e-23	0.982	9.76e-21
PBMC1 - Run 3	10X Genomics	0.993	4.39e-25	0.997	5.46e-30
PBMC1 - Run 4	10X Genomics	0.994	1.17e-25	0.994	3.35e-26
PBMC1 - Run 5	10X Genomics	0.994	2.48e-26	0.997	9.12e-29
PBMC1 - Run 6	10X Genomics	0.993	1.03e-29	0.993	2.47e-29
PBMC2 - Run A	10X Genomics	0.873	2.03e- 3	0.925	5.41e- 4
PBMC2 - Run B	10X Genomics	0.981	1.74e- 5	0.994	1.09e- 6
PBMC2 - Run C	10X Genomics	0.942	2.81e- 4	0.992	2.25e- 6
PBMC2 - all runs	10X Genomics	0.934	1.20e-12	0.971	3.94e-16
Lung	Drop-seq	0.990	6.34e-52	0.995	9.83e-60
Pancreas	Smart-seq2	0.980	8.25e-17	0.991	4.61e-20

Table S4. Evaluation of scPower gene expression prediction. We compare the observed number of expressed genes and the number estimated by scPower for each data set (separated by batches) and for different UMI expression cutoffs (>0 or >10), corresponding to Figure 2B-C,S6,S7,S18. The difference is quantified using r² and the corresponding p-values.

3. The usage of very light colors reduced the readability, such as light pink, light grey, light yellow in Fig 4 and 5.

We agree that the chosen colors were not optimal and updated our used color schemes.

4. Figure 2 used the same set of colored lines for different annotations in same Figure that often cause wrong interpretation of the data.

Thanks for pointing this out, we amended this issue in the updated figures.

Reviewer #3 (Expertise: scRNASeq analysis):

In this manuscript, Schmid et al. introduce scPower, a tool for power evaluation and sample size recommendation for single cell RNA-seq data, under the context of two-group differential expression (DE) tests for both DE genes and eQTLs. The authors adopted “pseudobulk” approach which is previously proposed by Dr. Mark Robinson, to calculate the detection power. However, the conclusion itself such as “the breadth is more preferable than the depth” is not unique.

Overall, the manuscript is well-written including the comprehensive literature reviews, and carries tests on different scenarios. However, I have some discretionary comments and critics for statistical modeling and assumptions.

Major:

1. The definition of detection power P_i is calculated as the product of expression probability (P_E) and detection power for DE (P_S). However, the author did not clarify why these two probabilities are independent. For example, simply imagine whether a gene expressed or not expressed follows a binomial distribution, then the statistical test will not become sensitive at the $P_E=0.5$. Another conclusion from the bulk RNA-seq analysis also validated in scRNA-seq is that DE genes with higher expression values are easier to be detected than the genes with lower expression values. Given the fact that P_E is a cell-wise probability quantifying expression rate larger than a threshold (e.g., 10 or 0), which is also related to the gene expression level; therefore, P_E is not independent from P_S . The author should properly justify the assumption of independence for P_E and P_S using real data.

We agree with the reviewer that P_E and P_S are not in general independent and our model is not based on this assumption. Instead, we assume conditional independence for P_E and P_S when conditioning on the mean (μ in Eq1) and dispersion (ϕ in Eq1) of the gene. See also equation (1) in the results (page 7, line 213-215):

$$\begin{aligned} P_i &= P(i \in E \wedge i \in S \mid n_s, n_c, r, \theta_e, \theta_p, \alpha) = \\ &= P(i \in E \mid n_s, \mu(n_c, r, \theta_e), \phi(n_c, r, \theta_e)) \cdot \\ &\quad P(i \in S \mid n_s, \mu(n_c, r, \theta_e), \phi(n_c, r, \theta_e), \theta_p, \alpha) \end{aligned}$$

2. Related to comment (1), the definition of P_E is similar to the definition of “detection rate” which is dependent on the sequencer platforms as discussed in [1], how to optimize the minimum expression threshold (δ) to calculate the P_E is still not clear. Using $\delta=10,3,0$ universally might not be reasonable. A data-driven approach could be a better choice.

We thank the reviewer for the interesting suggestion of using a data-driven approach to select the thresholds. In DEseq2, the filtering is optimized to maximize the number of significant findings (Bourgon, Gentleman, and Huber 2010). Adapting this data-driven approach, we tested which cutoffs will lead to the highest power, balancing the number of analyzed genes with the multiple testing burden. In our optimization, we consider both the individual-level UMI count threshold and the population level threshold. The function is added to our R package for the users (see also in the package vignette).

Using this new function, we evaluated the parameter combinations of Main Figure 3 with individual level UMI cutoffs from 0 to 20 and population level fraction of individual cutoffs from 0 to 1, for both Bonferroni and FDR correction (Figure 13). While the optimal UMI cutoff was 0 in all cases, the population level fraction of individuals cutoffs ranged between 0 and 0.4 with higher thresholds in case of Bonferroni, where the multiple testing burden is greater. Of note,

if a cutoff higher than 0 is chosen for the fraction of individuals cutoff, the power values for a threshold of 0 are only marginally smaller with a maximal difference of $< 0.1\%$. We visualize the trend exemplarily for the last column of plot Figure 13B, left panel: when comparing the maximal power for the different population cutoffs, the power stays nearly constant until it drops at some point (Figure 14). For this reason, we would suggest choosing a cutoff of 0 for both the UMI counts and the number of individuals or using our optimization function, if users are interested in the maximal power.

The user should however be aware that this will lead to higher false positive rates, as discussed in (Soneson and Robinson 2018). To what extent filtering of lowly expressed genes should be performed before the DE/eQTL analysis, is an open research question with contradicting results in different studies. For example, in two recent benchmarking studies for single cell DE analysis, one recommended to perform filtering of lowly expressed genes before (Soneson and Robinson 2018), while the other suggested no filtering (Vieth et al. 2019). To prevent false positives, best practice guidelines for differential gene expression with bulk RNA-seq recommend filtering of lowly expressed genes (SEQC/MAQC-III Consortium 2014) With our tool, we provide flexibility for the users to either choose the cutoff themselves according to their research question or use the function provided with our package.

Figure 13. Optimal UMI cutoff (A) and the population threshold (B) for expressed genes that maximize the detection power for each parameter combination, either using Bonferroni or FDR multiple testing correction. The shown example is the one of Main Figure 3A (a DE study with real world priors). A gene needs to be expressed in at least one individual ($>0\%$) with this UMI cutoff.

Figure 14. Relationship of the population cutoff and the detection power for different sample sizes (line colors) and 3000 cells per person with Bonferroni correction and UMI cutoff of 0. Same scenario as in Figure 13.

3. To my understanding, using this “pseudobulk” approach for power calculation can potentially bring batch effects for example each donor could possibly be defined as one batch (the sample variation can be confounded by batch effects). The author should address the concern of how to properly correct batches. Furthermore, Mark Robinson’s 2020 bioRxiv paper indicates a similar issue.

We would like to thank the reviewer for asking this important question (see also reviewer 1 comment 6). In general, the pseudobulk approach has proven to give more robust estimations compared to differential expression on single cell level, as it accounts better for the intrinsic variability of biological replicates and thus reduces the number of false discoveries (Crowell et al. 2020; Squair et al. 2021). The most important measure against batch effects is a proper experimental design. Genetic multiplexing (Kang 2018) and cell hashing (Stoeckius et al. 2018) are efficient ways to avoid having complete confounding of conditions and batches.

Still, correcting for batch effects between the donors is important. Equivalent to classical bulk DE and eQTL analysis, covariates can be added to the model of single cell RNA-seq analysis to account for different batches (W. Chen et al. 2020). This is even possible for unknown confounding variables based on latent factor analysis (Stegle et al. 2012; Buettner et al. 2017) and increases the power compared to non-batch corrected analysis (Hernández, Steyerberg, and Habbema 2004; Stegle et al. 2012; Kahan et al. 2014). To measure any reduction of power due to batch effects, we analyzed how well edgeR performs under batch effects of different strengths using powsimR and how much our scPower predictions deviated due to the batch effects.

We simulated the batch effect in powsimR by introducing different batch log fold changes to randomly sampled 20% of the genes, assigning the batches under the assumption of proper experimental design. This means that we generated 2 batches with 50% cases and 50% controls each. We observe that especially when the batch effects are higher than the effect sizes of the DE genes (mean effect size: absolute log₂ fold change of 2.8) the power is dropping drastically without batch correction in the covariates. As a side note, the differences

between scPower and powsimR in the condition without batch effects match the results shown in the main manuscript (Main Figure 3C-E and page 13-14, line 433-445) and can be explained by different modelling assumptions (muscat estimates are usually higher in these cases). Importantly, taking the batch as a covariate leads to the same power as in the simulation cases without any batch effects, even in cases with high batch effect sizes (Figure 15).

Therefore, we agree with the reviewer that batch effects are an important aspect to consider for the power analysis and we added a short section of this analysis to our manuscript (page 14, line 446-455). These results also show that our current analytic method is fully capable to deal with batch effects in properly designed experiments. We now emphasize in the manuscript that it is critically important to have a proper design of the experiment with non-confounded batches. This allows for a proper batch correction during the analysis and the power estimations with our method remain accurate.

Figure 15. PowsimR simulation results for number of expressed genes (A), DE power (B) and overall detection power (C) for data with batch effects of different log fold change (x-axis). The effect was evaluated for different parameter combinations of sample size - number of cells per sample (see facet header). The red lines visualize the prediction of scPower for this parameter combination. The power drops considerably if the log fold change of the batch effects is not corrected using covariates in the edgeR (violett), while with correction (green) it stays at the same level as without batch effects (blue).

4. The proposed method claims that using non-zero-inflation is sufficient to capture the pseudo bulk gene-wise distribution for different sequencers shown in (Table S4).

However, it has been reported that the raw read counts from Smart-seq have zero-inflation, while Unique Molecular Identifiers (UMIs) counts can be adequately modeled by a negative binomial distribution (NB) [2-4]. The manuscript should address this choice in light of this debate and argue for the appropriateness of their choice based on various datasets.

We fully agree with the reviewer that verifying the choice of the distribution for modeling the data is very important. As correctly pointed out, we shortly mention it in the third result section and in **Supplementary Table S5** (former **S4**). While we see the zero-inflation for Smart-seq data on single cell level (new **Supplementary Table S5** below) in line with other publications. Aggregation to pseudobulk counts removes it. The negative binomial distribution is preferred over a zero-inflated negative binomial for pseudobulk counts for more than 96% (Δ AIC < 10) / 91% (FDR(LRT) \geq 0.05) of all genes. We added the results on the single cell distributions to Supplementary Table S5 and expanded our explanation in the publication to clarify the argumentation (page 11, line 347-349).

We furthermore extended the number of analyzed data sets, as suggested by the reviewer. For the three previously analyzed data sets, we now show the results for all frequent cell types. Additionally, we analyzed two more Smart-seq2 data sets (Segal et al. 2019; Patil et al. 2018), as potential zero-inflation is especially an issue with Smart-seq2 data sets.

Technology (Tissue)	Cell type	Number cells	Mean UMI/read counts per cell	Number genes	Single cell counts		Pseudobulk counts	
					Δ AIC < 10	FDR(LRT) \geq 0.05	Δ AIC < 10	FDR(LRT) \geq 0.05
Smart-seq2 (Pancreas)	Alpha cells	998	1,061,059.9	17,715	69.4%	62.4%	99.4%	99.9%
	Ductal cells	389	1,226,815.0	16,108	67.9%	60.8%	98.5%	98.8%
	Beta cells	348	935,449.5	15,798	77.9%	71.5%	98.9%	100%
	Acinar cells	411	873,913.5	15,651	81.8%	76.1%	99.7%	100%
Smart-seq2 (fetal liver)	EpCAM+/NC AM+ cells	310	243,253.7	26,841	94.4%	93.2%	96.7%	91.4%
	EpCAM+/NC AM- cells	299	375,539.2	24,361	88.3%	85.4%	98.8%	99.5%
	EpCAM- cells	251	267,171.9	24,304	90.8%	88.3%	99.7%	100%
	CD235a-/CD45- cells	135	316,408.1	20,757	91.9%	89.6%	100%	100%
Smart-seq2 (T cells)	TEMRA cells	1234	668,510	24,829	99.8%	99.8%	99.7%	100%
	TEM cells	144	816,486.9	15,137	100%	100%	99.6%	100%

	TCM cells	143	837,545.3	15,114	100%	100%	99.5%	100%
10X (PBMCs)	CD4 T cells	2,755	4,865.5	13,537	100%	100%	100%	100%
	CD14+ Monocytes	1,162	4,425.5	12,057	100%	100%	100%	100%
	CD8 T cells	859	3,760.1	11,242	100%	100%	100%	100%
	NK cells	650	3,025.1	10,859	100%	100%	100%	100%
	B cells	410	3,913.6	9,780	100%	100%	100%	100%
	FCGR3A+ Monocytes	173	6,994.1	9,297	100%	100%	99.7%	100%
Drop-seq (Lung)	Macrophages	2,250	1,366.1	17,151	100%	100%	99.8%	100%
	B cell	1,181	1,099.6	15,356	100%	100%	100%	100%
	Type 2	988	1,343.1	14,545	100%	100%	100%	100%
	Secretory	598	861.2	12,660	100%	100%	99.9%	100%
	Ciliated	252	1,914.6	12,653	100%	100%	99.5%	100%
	T cell	781	747.3	12,606	100%	100%	100%	100%
	Transformed epithelium	502	1,104.8	12,604	100%	100%	100%	100%
	Mast cell	606	948.1	12,251	100%	100%	100%	100%
	Type 1	367	1,012.4	11,337	100%	100%	100%	100%
	Endothelium	333	713.6	9,688	100%	100%	100%	100%
	NK cells	338	615.0	8,842	100%	100%	100%	100%

Table S5. Evaluation of negative binomial (NB) versus zero-inflated negative binomial (ZINB) distribution for modelling of single cell counts (columns 6-7) and pseudo bulk counts (columns 8-9). Each gene with at least a count of 3 was evaluated, using both the Akaike Information Criterion (AIC) and the likelihood ratio test (LRT). Δ AIC was calculated as $AIC(NB) - AIC(ZINB)$, the threshold of 10 for Δ AIC to identify models with little support was chosen according to (Burnham, Anderson, and Huyvaert 2011). The p-values of the LRT were corrected for multiple testing using Benjamini Hochberg.

To explain our analyses in detail: we verified the distribution for our tested data sets by comparing the Akaike Information Criterion (AIC) of negative binomial fits and zero-inflated negative binomial fits (see updated **Supplementary Table S5** columns 6-7). The fits were performed for each cell type of the data set (except very rare cell types) and on all genes with

at least three raw counts. For the droplet-based data sets (the 10X PBMC data set and the drop-seq lung data set) nearly all genes were fitted better by the negative binomial distributions (lower AIC and high p-values in LRT). In line with (W. Chen et al. 2018; Svensson 2020) we observe that especially in the pancreas Smart-seq data set, about half of the genes are fitted better with a negative binomial distribution, indicating that zero inflation on a single cell level is more common in this technology.

However, we are not applying the negative binomial test (edgeR or DESeq) on the counts of the individual cells, but on the pseudo bulk counts per cell type. We evaluated whether the distributions of the pseudo bulk counts were better modelled by the negative binomial or the zero inflated negative binomial distribution. Updated **Supplementary Table S5** columns 8-9 show that for all technologies and the vast majority of genes (>96% / >91%) there is no significant evidence ($\Delta \text{AIC} < 10$ / $\text{FDR(LRT)} \geq 0.05$) of zero inflation in the pseudo bulk counts.

In conclusion, supported by both literature and our own observations, we are convinced that we accurately capture the excess of zeros in our negative binomial model and an explicit modelling of zero-inflation is not required.

5. It has been demonstrated that typical DE testing methodologies can suffer from spurious results that derive from clustering the data first and then running DE testing methods on the resulting clusters. The authors also fall into this camp that applying clustering first and then performing DE tests (lines 150-151). This will further hurt the multiple test adjustment.

We agree that double use of data for clustering and differential expression analysis might introduce biases, especially in the context of differential expression analysis between cell types (Lähnemann et al. 2020) within a sample. There, clusters are defined by distinct expression signatures, which would then also be picked up in differential gene expression analysis between cell types defined by these clusters. We argue that this issue is far less common in our setup, which compares expression levels of genes within the same cell type between individuals. Here, the clusters are not defined by expression signatures that distinguish individuals, but that distinguish cell types.

Our power analysis framework requires annotated single cell expression data, as other approaches (Crowell et al. 2020) do as well. A growing number of resources to create reliable annotations is available, providing marker genes and specific methods for transferring annotations (see for example all methods compared here (Abdelaal et al. 2019)). Cell type identification is also affected by experimental design choices and power estimation tools for cell type identification, such as (Heimberg et al. 2016; Abrams et al. 2019), can be combined with our power method to ensure a design that allows for proper cell type annotation.

*6. Followed by comment (5), the authors use two approaches FDR and FWER for DE/eQTL genes. [6.1] What does the distribution of the original p values obtained from the DE tests look like? [6.2] For FWER approach, why use threshold $0.05/(E*10)$? How many significant eQTL genes remain if using $E*100$, $E*1000$? [6.3] Also, for FDR approach, is there any advantages to adopt the Jung's formular and how it is related*

to your equation? From the description (line 877-880), the α' is an interval, then r_1 calculates the sum of marginal power based on different α' ? Why r_1 calculate the expected number of significant DE DEGs/eQTLs?

[6.1] Contrary to simulation tools, the output of our power analysis framework generates no p-values. Analytic power analysis compares the distributions of the test statistic under the null model and under the alternative model (e.g. applying a certain effect size). Based on the significance threshold the critical value of the test statistic is determined from the null distribution. Then the power is given by all the probability mass of the distribution under the alternative model that exceeds the critical value. The reviewer comment showed us that this was not described in sufficient detail, so we added additional explanations (page 10, line 325-329).

[6.2] The reviewer is right that we use a Bonferroni correction of $\alpha/(E*10)$ with $\alpha=0.05$ for the FWER approach in the eQTL setting (however the alpha can be readily changed from the default in the tool). With this, we follow the approach of the GTEx consortium (GTEx Consortium 2013), which assumes that for each gene on average 10 independent (uncorrelated) SNPs are tested in a genome-wide cis eQTL analysis. This leads to a total number of tests of E (=number of expressed genes) times the number of independent SNPs: $E*10$ and the Bonferroni adjusted P-value threshold is set at $\frac{0.05}{(E) * 10}$.

Following the reviewers suggestions, we made our tool more flexible by implementing the number of independent SNPs as an additional parameter for our tool. This is for example useful, when looking at different cis windows. When performing a cis eQTL analysis for a large cis region or a trans eQTL analysis, the estimate of 10 independent SNPs will be too low. We evaluated the effect for $E*100$ and $E*1000$, which resulted in a moderate decrease of the eQTL power compared to $E*10$ (Figure 16). This new parameter is now also available in our R package and on the website.

Figure 16: Evaluating the effect of more independent SNPs: the power for all parameter combinations of Main Figure 3B (each point representing one combination) was recalculated assuming instead of 10 independent SNPs (x-axis and default value in the paper) 100 or 1000 independent SNPs (y-axis, color-coded).

[6.3] We directly apply the formulas of Jung et al to estimate the FDR corrected threshold. The formula permits us to identify the specific raw p-value associated with a certain FDR cutoff. In contrast to the Bonferroni correction, which depends only on the number of tests, the FDR correction depends on the p-value distribution of all genes. As discussed in [6.1], the analytic method outputs the power without computing the p-values first. For this reason, we can not apply FDR correction directly. This makes it more difficult, however it is possible with the formula of Jung et al and numeric optimization. To explain the algorithm a bit more in detail: To achieve an FDR of α (in our case $\alpha = 0.05$), we need to identify the corresponding raw p-value cutoff α' (it is not an interval, but a single scalar value). We denoted this in the paper as $\alpha = FDR(\alpha')$. This means that we search for α' so that the false discovery rate is at most α when classifying each gene with a p-value smaller than α' as significantly differentially expressed (basically α is a q-value / FDR threshold while α' is the corresponding p-value threshold).

The FDR is the fraction of false positives among all rejected null hypotheses (predicted positives), which includes both the false positives and the true positives. Based on the probability integral transform, the distribution of p-values for the m_o true null hypotheses is uniform. Therefore, we get $m_o * \alpha'$ false positives at a raw p-value significance threshold of α' . m_o is here the number of expected expressed genes without the expected expressed DEGs/eQTLs, $(E) - (E_{DEG/eQTL})$. The expected number of true positives is directly derived from the power we reach for α' . Summing up the gene-wise power (at α') yields the expected number of significant DEGs/eQTLs $r_1(\alpha')$.

Using numerical optimization of the complete formula

$$FDR(\alpha') = \frac{m_o * \alpha'}{m_o * \alpha' + r_1(\alpha')}$$

with respect to the unknown parameter α' we identify the raw p-value threshold α' corresponding to the FDR threshold of α .

We updated the method description in this section to make the FDR correction clearer.

7. Since the effect size (lfc) or the cell-type-specific prior is computed from the pilot dataset, the authors should evaluate the similarity between the simulated data and the reference data in order to adopt the effect size for calling DE and ranking genes. Because the # of cells per cell type could impact the effect sizes. Specifically, the called DE gene ranks should be preserved like the real DE gene ranks. Moreover, since scPower uses effect size to determine DE genes, it is not fair to compare with powsimR and muscat with respect to running time for they use different ways to call DE genes in the Figure3.

As mentioned in comment 6, we want to emphasize that we do not simulate the count data itself, as done by powsimR and muscat, but use an analytic strategy. For all analyses of the main text (except Main Figure 5A,B) the effect sizes and the DE gene expression ranks are directly taken as priors from a reference data set as input to the power estimation. Therefore they match exactly with the reference data. In these analytic calculations, the expression priors combined with the ranks are used to define the shape of the gene-wise expression distribution and the effect size priors are used to define the shift between null and alternative models.

The only part that we simulate in the DE analysis are the effect sizes and expression ranks of the prototypic scenarios (old manuscript: Figure 4A,B and 5A,B, new manuscript: Figure 5A,B and Figure S20A,B). The goal of the prototypic scenarios was to generate priors that have a similar distribution to the priors observed in the real world data set, but to represent extreme

cases (high or low effect sizes and high expression ranks or uniformly distributed expression ranks). To prove this, we plotted the expression ranks and effect sizes, and show that the simulated distributions in red and orange frame the different real world examples in different grey shades (Figure 17). With these extreme priors, we visualized the effect of the priors on the optimal parameter combinations, such as a higher required sample size for low effect sizes.

Regarding the concern that the number of cells per cell type could impact the effect sizes: the effect sizes in our model represent differences between samples on the individual-level. A lower number of cells will lead to lower mean values in the pseudobulk and consequently this reduces the DE power. However, the effect sizes themselves are not changed by the number of cells. This is also in line with the idea that the effect sizes represent the true differences between the two groups, which are completely independent of the used experimental parameters such as sample size and number of cells. The experimental parameters might make it more difficult to find significant differences, e.g. a small sample size or low numbers of cells lead to a reduced power to detect an effect of a specific size, but they don't change the effect size.

Also muscat and powsimR work according to these principles. The effect sizes are given as priors, which shift the negative binomial mean of the second groups (for muscat also other variants exist, but we focus on this scenario). The simulated counts are then drawn for the respective negative binomial distribution of the group.

We disagree that the comparison with muscat and powsimR is not fair. In all cases, the simulations and our analytic model, the power is calculated based on effect sizes, multiple-testing corrected significance threshold and the experimental parameters. As Figure 3 shows, the estimated powers are comparable between the methods. Therefore, we argue that the methods can be used interchangeably, when the goal is to find the optimal design for a single cell multi-sample DE analysis. The runtime and memory comparison shows that an evaluation of many different designs is not feasible with the simulation-based models and so neither is budget optimization of the experimental parameters. Hence, our analytic method is an important alternative to the simulations. In contrast, we of course want to acknowledge that for other use cases the simulation-based methods are needed, such as benchmarking of different normalizations and DE methods.

Figure 17. Comparison of simulated priors from prototypic scenarios (red and orange) with priors from real world studies (grey shades), for the expression ranks (A), the absolute log fold change (B) and the R-squared values (C). Simulated distributions frame real word examples.

8. There is only a small section talking about rare cell type detection, and it seems independent from the overall manuscript because it is not related to the theme of detection power.

We agree that the section “Power to detect rare cell types” was out of place and that the connection with the detection power did not become clear. The main theme of our manuscript,

the detection power, depends on the prerequisite that a sufficient number of cells is detected for each individual. For completeness, we added this section to show that this prerequisite can be evaluated by calculating the cell type detection power. This power is additionally covered on our website. We hope that by shortening and moving the section to the beginning of the results and explaining that it is a prerequisite we solved this issue (page 6, line 182-193).

9. There is an assumption made in the model that cell types present, and their proportion in the sample will be replicated in future experiments (otherwise the power calculation is not appropriate). This is a very strong assumption, especially considering that cell types are inferred from clustering, which can be affected by variability in the composition, depends on the resolution used to determine clusters etc. The manuscript should include (1) an analysis of the robustness of the method as proposed to variability in cell composition; (2) if necessary, an extension to the proposed method to address this.

We agree with the reviewer that the cell type proportions will vary both between the experiments and between individuals within one experiment. Regarding variation between experiments, we want to emphasize that the prior knowledge ideally comes from a pilot experiment. If experimental and analysis parameters (such as the resolution during the clustering) are held constant, they are expected to provide a good estimate of the cell type proportions. We evaluated for the two PBMC data sets from Main Figure 2, how much variation exists between individuals and data sets. The first data set, PBMC1, was generated by us, while the second data set, PBMC2, is from a published study (Kang et al. 2018). As shown in Figure 18, there is moderate variation across the individuals and data sets with a standard deviation of at most 0.14 between individuals of one data set (for the CD4 T cells in PBMC2 data set) and maximal difference of median cell type frequency of 0.06 (again for the CD4 T cells). Interestingly, we see very high concurrence of the cell proportions between the data sets despite the fact that the cell types in the two datasets were annotated using different methodologies, highlighting the robustness to differences in clustering. In case of doubt we suggest taking a lower bound estimation of the cell type frequency, which will only lead to a more conservative estimate for the power and the optimal parameters.

Figure 18: Cell type frequency across individuals in two different PBMC data set (PBMC1: 14 individuals, PBMC2: 8 individuals)

Regarding variation between individuals within a cohort, we performed an additional evaluation using muscat to measure any decrease in power. We simulated different cell type compositions in both DE groups (Figure 19). We applied scPower with two different cell type frequencies, once with the overall target cell type frequency (default approach in dark orange) and once with the cell type frequency of the smaller group 1 (conservative approach in light orange).

Figure 19. Muscat simulation results for number of expressed genes (A), DE power (B) and overall detection power (C) for imbalanced cell proportions in both groups (x-axis). The cell proportion here represents the fraction of all measured cells of a specific cell type that fall into group 1 (e.g. a value of 0.3 means that 30% of measured cells belong to group 1 and 70% to group 2). The effect was evaluated for different parameter combinations of sample size - number of cells per sample (see facet header). For small sample sizes the power estimated with muscat (blue) is lower than the default scPower estimation that assumes balanced groups with each the target cell type frequency (dark orange). However, a conservative estimation of scPower can be reached by scaling the cell type frequency by the cell proportion of group 1 (light orange). This represents a good lower bound power estimation for scenarios with imbalanced cell proportions between both groups.

The expression probability is not affected by imbalanced cell type compositions if we choose the cutoff of at least one count in at least one individual. For more than one count in at least one individual, the expression probability increases even slightly in imbalanced designs. So as long as moderate cutoffs are chosen - which we strongly recommend - the expression probability is not affected.

In contrast, the DE power and due to that also the overall detection power decreases slightly in imbalanced designs compared to the default scPower estimation. However, even for imbalanced designs, the decrease is modest, especially if enough samples and cells per sample are measured (Figure 19). For a sample size of 16 and 3,000 cells, nearly no effect was visible even for drastically imbalanced groups, with only 10% of all cells in group 1. For each parameter combination, the median power decreased only slightly from the 50% balanced design to the 20% imbalanced design .

In contrast, the conservative scPower estimate provides a good lower bound estimation in case of imbalanced cell type proportions. For this reason, we recommend that especially in the case of drastically imbalanced groups and small sample sizes to estimate the cell type frequency based on the values of the lower group. We stated this now also in a section of our manuscript (page 14, line 456-459).

Related to this issue, a recent benchmarking study on single-cell eQTL workflows (Cuomo et al. 2021) suggested adding the number of cells as a covariate to increase the power. This approach could additionally lead to less decrease of power and so a better agreement with our estimations.

10. Since the authors mentioned the feasibility of extending the two-group test into multi-group test via this pseudobulk approach, it is better to showcase the investigation result on one of the PBMC datasets.

We thank the reviewer for the suggestion. Following the advice, we now show the combination of our method with the approach of (Lyles, Lin, and Williamson 2007) in an additional vignette in our package (<https://github.com/heiniglab/scPower>). In an example, we evaluated the power for a three group comparison with different sample sizes (Figure 20).

Figure 20. Power estimation for a three group comparison for different sample sizes (example from vignette “Extending scPower to complex designs”).

The method of (Lyles, Lin, and Williamson 2007) is a general method for power analysis of generalized linear models (GLMs). Therefore, it enables multi-group tests and also other more complex analyses that are based on GLMs. It requires to simulate one example data set and perform a GLM analysis to get the corresponding test statistics from it. With this, the power for different sample sizes can be evaluated. It does not require repeating the simulation multiple times, making it much more efficient in terms of runtime compared to purely simulation based approaches. Combining it with our scPower method, a large number of complex designs can be evaluated beyond the two-group comparison with still relatively moderate runtime and memory requirements. Please have a look at the vignette for more details.

To make sure that the estimates are accurate with the new model, we also performed one evaluation based on the two group comparison. The estimations of the general GLM method matched closely the estimations of (Zhu and Lakkis 2014) - the default method for two-group comparisons in scPower - that was specifically designed for the two group comparison (Figure 21).

Figure 21. Power calculation for a two group comparison using either the scPower default method or the complex design formula. Both estimates match closely for different log2 fold changes (x axis) and sample sizes (panel header).

Minor:

11. In line 1016, n_s is the quotient of two quantities.

We thank the reviewer for looking carefully through the methods section. In our implementation, we always round the quotient down to make sure we get an integer for the sample size in the end. We corrected the formula to display this (see section “Cost calculation and parameter optimization for a given budget” on page 32, line 1161):

$$n_s = \text{floor}\left(C_t / \left(\frac{C_k}{6 * n_{s,l}} + \frac{n_c * r * C_f}{r_f}\right)\right)$$

12. Some R package names are in italic font some are not.

We thank the reviewer for highlighting this inconsistency and now formatted all R packages in italic.

13. scPower calculates the gene-wise means and use splatter mixture model to capture the cell-wise distribution of the means corresponding to all the genes. Why not directly use splatter mixture model to capture the pseudobulk counts per cell type? Is there a merit of applying NB first?

The key reason for first modeling the count distributions on the single cell level is that it allows us to introduce the number of cells as a parameter of the pseudo-bulk distribution. This enables our model to compute power for varying cell numbers. Modeling the pseudo bulk directly with splatter would prohibit us from introducing the number of cells as a parameter of the expression distribution.

Regarding the negative binomial fit, this is analogous to how it is done in splatter. Also in splatter, first the mean and dispersion parameter of each gene is estimated using a negative binomial model. Then the means over all genes are fitted using the splatter mixture model and the dispersions as a function dependent on the means.

We are happy to explain further details of our model compared to splatter for the reviewer. While it is true that our model is based on the splatter model, we added some important changes that improved our fit. Splatter models the distribution only for genes found as expressed in the pilot data set. However, this leads to the restriction that the number of expressed genes can never be higher than the number of expressed genes in the pilot data set used for fitting the distribution. In reality, a higher number of cells per sample can lead to a higher number of expressed genes than observed in the pilot data set. To handle this, we fitted a left-censored distribution with the censoring point of $1/\#cells$, which is the lowest mean expression level that can be captured in a data set. With this method, we can also estimate the number of expressed genes correctly for data sets with more cells than in the pilot data set.

Furthermore, we want to clarify that the parameterization of the mixture model by the number of reads is also unique to our model and not part of splatter. This is a crucial step for evaluating different read depths.

Reference:

*[1] Hicks, Stephanie C., et al. "Missing data and technical variability in single-cell RNA-sequencing experiments." *Biostatistics* 19.4 (2018): 562-578.*

*[2] Grün, Dominic, Lennart Kester, and Alexander Van Oudenaarden. "Validation of noise models for single-cell transcriptomics." *Nature methods* 11.6 (2014): 637-640.*

*[3] Svensson, Valentine. "Droplet scRNA-seq is not zero-inflated." *Nature Biotechnology* 38.2 (2020): 147-150.*

*[4] Townes, F. William, et al. "Feature selection and dimension reduction for single-cell RNA-Seq based on a multinomial model." *Genome biology* 20.1 (2019): 1-16.*

References

Abdelaal, Tamim, Lieke Michielsen, Davy Cats, Dylan Hoogduin, Hailiang Mei, Marcel J. T. Reinders, and Ahmed Mahfouz. 2019. "A Comparison of Automatic Cell Identification

- Methods for Single-Cell RNA Sequencing Data." *Genome Biology* 20 (1): 194.
- Abrams, Douglas, Parveen Kumar, R. Krishna Murthy Karuturi, and Joshy George. 2019. "A Computational Method to Aid the Design and Analysis of Single Cell RNA-Seq Experiments for Cell Type Identification." *BMC Bioinformatics* 20 (Suppl 11): 275.
- Bourgon, Richard, Robert Gentleman, and Wolfgang Huber. 2010. "Independent Filtering Increases Detection Power for High-Throughput Experiments." *Proceedings of the National Academy of Sciences of the United States of America* 107 (21): 9546–51.
- Buettner, Florian, Naruemon Pratanwanich, Davis J. McCarthy, John C. Marioni, and Oliver Stegle. 2017. "F-scLVM: Scalable and Versatile Factor Analysis for Single-Cell RNA-Seq." *Genome Biology* 18 (1): 212.
- Burnham, Kenneth P., David R. Anderson, and Kathryn P. Huyvaert. 2011. "AIC Model Selection and Multimodel Inference in Behavioral Ecology: Some Background, Observations, and Comparisons." *Behavioral Ecology and Sociobiology* 65 (1): 23–35.
- Chen, Wenan, Yan Li, John Easton, David Finkelstein, Gang Wu, and Xiang Chen. 2018. "UMI-Count Modeling and Differential Expression Analysis for Single-Cell RNA Sequencing." *Genome Biology* 19 (1): 70.
- Chen, Wenan, Silu Zhang, Justin Williams, Bensheng Ju, Bridget Shaner, John Easton, Gang Wu, and Xiang Chen. 2020. "A Comparison of Methods Accounting for Batch Effects in Differential Expression Analysis of UMI Count Based Single Cell RNA Sequencing." *Computational and Structural Biotechnology Journal* 18 (March): 861–73.
- Chen, Yunshun, Aaron T. L. Lun, Davis J. McCarthy, Matthew E. Ritchie, Belinda Phipson, Yifang Hu, Xiaobei Zhou, Mark D. Robinson, and Gordon K. Smyth. 2021. "FilterByExpr: Filter Genes by Expression Level in edgeR: Empirical Analysis of Digital Gene Expression Data in R." January 14, 2021. <https://rdrr.io/bioc/edgeR/man/filterByExpr.html>.
- Chen, Yunshun, Aaron T. L. Lun, and Gordon K. Smyth. 2016. "From Reads to Genes to Pathways: Differential Expression Analysis of RNA-Seq Experiments Using Rsubread and the edgeR Quasi-Likelihood Pipeline." *F1000Research* 5 (June): 1438.
- Crowell, Helena L., Charlotte Soneson, Pierre-Luc Germain, Daniela Calini, Ludovic Collin, Catarina Raposo, Dheeraj Malhotra, and Mark D. Robinson. 2020. "Muscat Detects Subpopulation-Specific State Transitions from Multi-Sample Multi-Condition Single-Cell Transcriptomics Data." *Nature Communications* 11 (1): 6077.
- Cuomo, Anna S. E., Giordano Alvani, Christina B. Azodi, single-cell eQTLGen consortium, Davis J. McCarthy, and Marc Jan Bonder. 2021. "Optimising Expression Quantitative Trait Locus Mapping Workflows for Single-Cell Studies." *bioRxiv*. <https://doi.org/10.1101/2021.01.20.427401>.
- GTEX Consortium. 2013. "The Genotype-Tissue Expression (GTEx) Project." *Nature Genetics* 45 (6): 580–85.
- Heimberg, Graham, Rajat Bhatnagar, Hana El-Samad, and Matt Thomson. 2016. "Low Dimensionality in Gene Expression Data Enables the Accurate Extraction of Transcriptional Programs from Shallow Sequencing." *Cell Systems* 2 (4): 239–50.
- Hernández, Adrián V., Ewout W. Steyerberg, and J. Dik F. Habbema. 2004. "Covariate Adjustment in Randomized Controlled Trials with Dichotomous Outcomes Increases Statistical Power and Reduces Sample Size Requirements." *Journal of Clinical Epidemiology* 57 (5): 454–60.
- Johnson, W. Evan, Cheng Li, and Ariel Rabinovic. 2007. "Adjusting Batch Effects in Microarray Expression Data Using Empirical Bayes Methods." *Biostatistics* 8 (1): 118–27.
- Kahan, Brennan C., Vipul Jairath, Caroline J. Doré, and Tim P. Morris. 2014. "The Risks and Rewards of Covariate Adjustment in Randomized Trials: An Assessment of 12 Outcomes from 8 Studies." *Trials* 15 (April): 139.
- Kang, Hyun Min, Meena Subramaniam, Sasha Targ, Michelle Nguyen, Lenka Maliskova, Elizabeth McCarthy, Eunice Wan, et al. 2018. "Multiplexed Droplet Single-Cell RNA-Sequencing Using Natural Genetic Variation." *Nature Biotechnology* 36 (1): 89–94.
- Lähnemann, David, Johannes Köster, Ewa Szczurek, Davis J. McCarthy, Stephanie C.

- Hicks, Mark D. Robinson, Catalina A. Vallejos, et al. 2020. "Eleven Grand Challenges in Single-Cell Data Science." *Genome Biology* 21 (1): 31.
- Law, Charity W., Yunshun Chen, Wei Shi, and Gordon K. Smyth. 2014. "Voom: Precision Weights Unlock Linear Model Analysis Tools for RNA-Seq Read Counts." *Genome Biology* 15 (2): R29.
- Lyles, Robert H., Hung-Mo Lin, and John M. Williamson. 2007. "A Practical Approach to Computing Power for Generalized Linear Models with Nominal, Count, or Ordinal Responses." *Statistics in Medicine*. <https://doi.org/10.1002/sim.2617>.
- Mandric, Igor, Tommer Schwarz, Arunabha Majumdar, Kangcheng Hou, Leah Briscoe, Richard Perez, Meena Subramaniam, et al. 2020. "Optimized Design of Single-Cell RNA Sequencing Experiments for Cell-Type-Specific eQTL Analysis." *Nature Communications* 11 (1): 5504.
- Patil, Veena S., Ariel Madrigal, Benjamin J. Schmiedel, James Clarke, Patrick O'Rourke, Aruna D. de Silva, Eva Harris, et al. 2018. "Precursors of Human CD4 Cytotoxic T Lymphocytes Identified by Single-Cell Transcriptome Analysis." *Science Immunology* 3 (19). <https://doi.org/10.1126/sciimmunol.aan8664>.
- Segal, Joe M., Deniz Kent, Daniel J. Wesche, Soon Seng Ng, Maria Serra, Bénédicte Oulès, Gozde Kar, et al. 2019. "Single Cell Analysis of Human Foetal Liver Captures the Transcriptional Profile of Hepatobiliary Hybrid Progenitors." *Nature Communications* 10 (1): 3350.
- SEQC/MAQC-III Consortium. 2014. "A Comprehensive Assessment of RNA-Seq Accuracy, Reproducibility and Information Content by the Sequencing Quality Control Consortium." *Nature Biotechnology* 32 (9): 903–14.
- Soneson, Charlotte, and Mark D. Robinson. 2018. "Bias, Robustness and Scalability in Single-Cell Differential Expression Analysis." *Nature Methods* 15 (4): 255–61.
- Squair, J. W., M. Gautier, C. Kathe, and M. A. Anderson. 2021. "Confronting False Discoveries in Single-Cell Differential Expression." *bioRxiv*. <https://doi.org/10.1101/2021.03.12.435024>.
- Stegle, Oliver, Leopold Parts, Matias Piipari, John Winn, and Richard Durbin. 2012. "Using Probabilistic Estimation of Expression Residuals (PEER) to Obtain Increased Power and Interpretability of Gene Expression Analyses." *Nature Protocols* 7 (3): 500–507.
- Svensson, Valentine. 2020. "Droplet scRNA-Seq Is Not Zero-Inflated." *Nature Biotechnology* 38 (2): 147–50.
- Vieth, Beate, Swati Parekh, Christoph Ziegenhain, Wolfgang Enard, and Ines Hellmann. 2019. "A Systematic Evaluation of Single Cell RNA-Seq Analysis Pipelines." *Nature Communications* 10 (1): 1–11.
- Zhu, Haiyuan, and Hassan Lakkis. 2014. "Sample Size Calculation for Comparing Two Negative Binomial Rates." *Statistics in Medicine* 33 (3): 376–87.

Reviewers' Comments:

Reviewer #1:

Remarks to the Author:

The authors have now clarified all of the issues that I have raised in my prior comments. Now both the DE and the eQTL parts are better explained and the authors incorporated several small, but important improvements, for example an adaptive gene detection threshold for eQTLs. Furthermore, limitations such as batch effects are discussed, so that the user is aware.

Most importantly, the confusion about power varying with reads vs. sample size was clarified. With the new bubble plot it is now clear that also the number of cells in this analysis was also variable.

Now the manuscript seems ready for publication.

Reviewer #2:

Remarks to the Author:

This revision has major improvement comparing the original version. Most of my comments were appropriately address. I have only one minor comment, i.e. new Figure 1 is a very brief overview. I would suggest moving it to supplemental.

Reviewer #3:

Remarks to the Author:

Thanks for the authors' hard work in revising the manuscript which is now significantly improved compared to the first edition. The extensive simulation seems convincing; However, I still have one remaining concern.

Regarding optimizing the minimum expression threshold (δ), it is great that the author added one equation for the package. However, it is a heuristic solution because the users need to preselect some values and decide which one is the best solution. Is it also a convex optimization process?

Response to reviewer comments

Reviewer #1 (Remarks to the Author):

The authors have now clarified all of the issues that I have raised in my prior comments. Now both the DE and the eQTL parts are better explained and the authors incorporated several small, but important improvements, for example an adaptive gene detection threshold for eQTLs. Furthermore, limitations such as batch effects are discussed, so that the user is aware. Most importantly, the confusion about power varying with reads vs. sample size was clarified. With the new bubble plot it is now clear that also the number of cells in this analysis was also variable.

Now the manuscript seems ready for publication.

We thank the reviewer for this feedback.

Reviewer #2 (Remarks to the Author):

This revision has major improvement comparing the original version. Most of my comments were appropriately address. I have only one minor comment, i.e. new Figure 1 is a very brief overview. I would suggest moving it to supplemental.

We thank the reviewer for this feedback. We agree that Figure 1 is only a brief overview of the method. However, a general graphical overview of the methods is usually very helpful for users to understand the approach and we decided to keep it on a high-level to also allow users with less statistical knowledge to follow the basic algorithm. For this reason, we would prefer to keep Figure 1 in the main text.

Reviewer #3 (Remarks to the Author):

Thanks for the authors' hard work in revising the manuscript which is now significantly improved compared to the first edition. The extensive simulation seems convincing; However, I still have one remaining concern.

Regarding optimizing the minimum expression threshold (δ), it is great that the author added one equation for the package. However, it is a heuristic solution because the users need to preselect some values and decide which one is the best solution. Is it also a convex optimization process?

We thank the reviewer for this feedback. The idea of a convex optimization process is interesting. However, we are convinced that the heuristic solution is fully sufficient for our application, as the search space of possible cutoff values is discrete and very small (as discussed the last response, large UMI cutoffs are not sensible, as the number of quantifiable genes will drop to low and this will always reduce the overall power). In combination with our efficient analytic solution, the cutoff optimization is in practice fast enough and does not constitute a bottleneck that would prevent interesting additional applications of our method.